



# Deducing Land-Atmosphere Coupling Regimes from SMAP Soil Moisture

Payal R. Makhasana[1*], Joseph A. Santanello[2], Patricia M. Lawston-Parker[2,3], Joshua K. Roundy[1]

[1]Department of Civil, Environmental, and Architectural Engineering, University of Kansas, Lawrence, Kansas
[2]Hydrological Sciences Laboratory, NASA Goddard Space Flight Center, Greenbelt, Maryland
[3]Earth System Science Interdisciplinary Center, University of Maryland, College Park, Maryland

*Correspondence to*: Payal R. Makhasana (prmakhasana@ku.edu)

**Abstract.** In recent years, there has been a growing recognition of the significance of Land-Atmosphere (L-A) interactions and feedback mechanisms and their importance for weather and climate prediction. Soil moisture plays a critical role in
mediating L-A interactions; therefore, this research assesses the impact of different soil moisture datasets on the classification and distribution of L-A coupling regimes. Using SMAP Level 3 (SMAPL3) and SMAP Level 4 (SMAPL4) soil moisture data, we examine the persistence of dry and wet coupling regimes over two decades (2003-2022), exploring how soil moisture influences coupling classification. An inherent challenge in assessing the significance of soil moisture in L-A coupling classification lies in the need for consistent and unbiased observations of the atmospheric state, represented through metrics
such as Convective Triggering Potential (CTP), offering insights into atmospheric stability and Humidity Index (HI), which quantifies moisture within the atmosphere. The study utilizes a Triple Collocation-based merging process to address this issue and combines three reanalysis datasets for CTP and HI. Despite significant correlated errors within the individual reanalysis datasets, the merged product demonstrates enhanced performance, showcasing increased accuracy in capturing atmospheric conditions. When combined with the merged CTP and HI for coupling classification, a higher lag-correlation between soil
moisture and the CTP-HI metrics contribute to the persist coupling behaviour, potentially suggesting that temporal consistency is a leading factor. SMAPL4 demonstrates stronger persistence of the wet and dry coupling regimes as compared to SMAPL3. The stronger persistence is partially due to the higher observation count, though it may partially be linked to the unique characteristics of the SMAPL4's assimilation process. This suggests that SMAPL4's approach may offer a robust approximation when assessing land-atmosphere interactions, highlighting the inherent differences between SMAPL3 and
SMAPL4 datasets. These findings lay the groundwork for understanding the sensitivity of drought evolution to soil moisture variations by gaining insight into the quantification of coupling strength, thereby providing critical insights for future drought modelling and prediction efforts.

**Key Word.** Land-atmosphere coupling, SMAP observation, Triple collocation, CTP-HI framework.



## 1 Introduction

Land-atmosphere (L-A) interactions are critical to Earth's complex climate system processes and environmental sustainability (Seo and Dirmeyer, 2022; Seneviratne and Stöckli, 2008). These interactions are primarily driven by the two-way energy, momentum, and mass exchanges between the land surface and the overlying atmosphere (Hsu and Dirmeyer, 2023). Among the many components influencing L-A interactions, soil moisture is a critical element (Zhou et al., 2019; Santanello et al., 2018; Saini et al., 2016; Alexander et al., 2022; Wakefield et al., 2019 ; Findell et al., 2023).  Soil moisture is not merely a

passive participant but the foundation and active modifier of water and heat transfer between the land and the atmosphere (Qi et al., 2023). It serves as a vital component in the climate system and represents an essential climate variable (ECV).

Soil moisture is a critical intermediary in L-A feedback loops, affecting a wide range of atmospheric processes at local and regional scales (Seo and Ha, 2022). Its influence on partitioning energy at the land surface into sensible and latent heat fluxes directly impacts weather patterns, climate variability, and extreme meteorological events (Zhou et al., 2019). L-A interactions

are often separated into different coupling regimes (Bennet et al., 2023), such as dry and wet coupling regimes. Dry coupling refers to conditions where the land's dryness limits moisture availability to the atmosphere, typically resulting in less cloud formation and precipitation, leading to hotter, drier conditions that exacerbate droughts. Conversely, wet coupling describes a scenario where abundant soil moisture enhances evaporation and transpiration, increasing humidity, cloud formation, and potentially more precipitation. Understanding these interactions is crucial for comprehending how the land's condition affects

the atmosphere and vice versa to predict weather patterns, manage ecological systems, and prepare for short-term climate change impacts.

Classifying L-A coupling regimes requires incorporating measures of both the land state and the atmosphere to illuminate the L-A coupling. An integrated approach, highlighted by Findell and Eltahir (2003), offers insight into the complex exchanges and feedback loops between the Earth's surface and the air above. Specifically, Findell and Eltahir used the Convective

Triggering Potential (CTP) and Humidity Index (HI) as metrics to classify L-A feedback based on the likelihood of convective precipitation in an observationally driven modelling framework. Within this framework, CTP is critical in gauging the potential for convection formation by assessing atmospheric stability, while HI measures low atmospheric moisture levels. The CTP-HI framework, was further developed by Roundy et al. (2013), which evaluates morning atmospheric conditions via vertical temperature and humidity profiles, linking the thermodynamic preconditioning of the lower troposphere to soil moisture levels

that could trigger or suppress convective activities. This method provides a sophisticated tool for understanding and predicting weather patterns and regimes based on L-A interactions.

Roundy et al. (2013) introduced an approach to L-A interaction classification by leveraging soil moisture observations directly within the CTP and HI space. This method underscores the critical role of soil moisture as an indicator of surface conditions and a crucial driver influencing convective dynamics. The unique aspect of integrating soil moisture into the CTP-HI

framework provides an observational based approach for understanding L-A coupling. Yet, the extent of soil moisture's





sensitivity within this classification system remains largely unexplored, especially relative to satellite-based observations of soil moisture.

Furthermore, quantifying soil moisture sensitivity within the CTP-HI framework requires a consistent and reliable data set of atmospheric profiles (i.e., for CTP and HI) to isolate the soil moisture sensitivity within L-A coupling. However, this is not straightforward, as global observations of lower tropospheric temperature and humidity are inherently limited (Teixeira et al., 2021). Reanalysis data provide the best global estimate of atmospheric data for quantifying the CTP-HI, yet inherent uncertainties in the data cannot be overlooked. These uncertainties often arise from varied data sources, measurement approaches, and spatial and temporal resolution. These factors, along with diverse data assimilation techniques, can contribute to biases and uncertainties within climate reanalysis, thus complicating the quantification of land-atmosphere dynamics, as highlighted by both Jach et al. (2022) and Mukherjee and Mishra (2022).

One way to address the challenge of uncertainties in reanalysis data sets, is to employ data merging techniques, as suggested by Sun and Fu (2021), Lu et al. (2021), and Feng and Wang (2018), to create a more reliable data set of CTP and HI based on multiple reanalysis data. Many merging techniques exist, including M-kernel merging (Zhou et al., 2003), optimal interpolation (Lorenzo et al., 2017), Random Forest algorithms (Nguyen et al., 2021), and approaches rooted in Bayesian analysis (Wilson and Fronczyk, 2016) to name a few. However, the Triple Collocation (TC) method has emerged as an invaluable technique for estimating error variances within datasets, as evidenced by research from Peng et al. (2021), Dash and Sinha (2022), and Saha et al. (2020). Its widespread application is primarily due to its effectiveness in using the statistical attributes of multiple datasets to discern the accurate signal and quantify the uncertainties associated with each dataset. Therefore, using the TC method to merge reanalysis data sets of CTP and HI based has the potential to provide a more robust analysis of land-atmosphere interactions with which to assess the impact of soil moisture on the strength of the coupling regimes.

This study aims to enhance our understanding of L-A coupling classification by developing and applying merged CTP and HI datasets to isolating the impact of different soil moisture datasets on the CTP-HI framework and the associated coupling strength. Specifically, it focuses on the L-A coupling outcome from the Soil Moisture Active Passive Level 3 (SMAPL3) satellite remote sensing (Entekhabi et al., 2016) and the assimilation based SMAP Level 4 (SMAPL4) soil moisture product that synthesizes remote sensing data with land surface modelling (Rolf et al., 2017). The goal of this comparative study is to uncover how soil moisture, as detected through direct satellite observations and assimilated data products, influences L-A coupling strength across the globe. The anticipated insights into how soil moisture variability influences coupling strength will provide critical advancements for assessing hydrological extremes. The comprehensive analysis of the coupling series, detailed in the results and discussion section, underscores the study's contribution towards improving predictive models for weather and climate phenomena.



## 2 Methodology

### 2.1 L-A Coupling Classification

To quantify the L-A coupling, the CTP-HI framework is used. The CTP-HI framework is based on the work of Findell and Eltahir (2003) and was revised by Roundy et al. (2013) to be an observational-based framework. This framework utilizes 95 measures of atmospheric stability (CTP), a measure of atmospheric humidity (HI) and soil moisture to classify the CTP-HI space into coupling regimes as shown in Fig. 1.

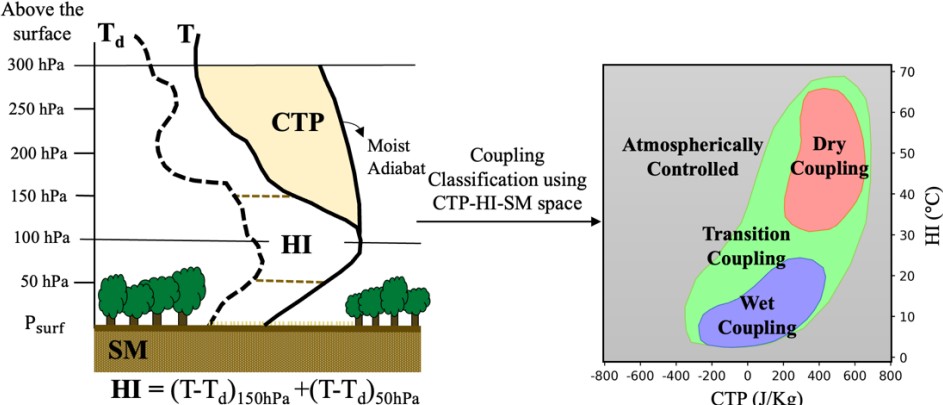

**Figure 1: A visual representation of the Soil Moisture (SM), Convective Triggering Potential (CTP), and Humidity Index (HI) on a thermodynamic diagram, along with a conceptualized representation of coupling classification regimes, "(adapted from Roundy et 100 al. (2013)".**

The CTP is the integrated area between the temperature profile and a moist adiabat between 100 and 300 hPa above the surface. It quantifies the stability of the atmosphere, where a more negative value indicates a stable condition, and a more positive value indicates unstable conditions. The HI quantifies moisture content in the lower atmosphere, explicitly reflecting the low-level dew point depression. The HI is calculated as the sum of the dew point depressions at 50 and 150 hPa above the surface. 105 A large value of HI indicates a dry atmosphere as there is a significant difference between the temperature and the dew point temperature at the specified pressure levels, suggesting low moisture content. As the dew point temperature approaches the temperature, the HI decreases and reaches zero when the atmosphere is saturated.

In the revised CTP-HI framework developed by Roundy et al. (2013), the interplay between soil moisture and atmospheric conditions is distinguished into four specific coupling regimes: wet coupling, dry coupling, transitional, and atmospherically 110 controlled; and summarize the complex relationship between soil moisture content and the feedback from the land to the atmosphere in a generalized context. The regimes are identified by analysing the distribution of soil moisture in the joint CTP-HI-SM space using the two-sample Kolmogorov-Smirnov test between a sub-space marginal distribution and the climatological distribution of soil moisture. This process is repeated using multiple thresholds and sampling resolutions within the CTP-HI space to arrive at a robust classification considering algorithmic parameter uncertainty (see Roundy et. al 2013 for





more details). Wet coupling is identified as regions in the CTP-HI space with predominantly wetter soils and is indicative of increased latent heat fluxes conducive to moist conditions that can initiate convection, leading to a positive feedback loop that amplifies convection, increases moist static energy, and lowers the lifting condensation level. On the other hand, the dry coupling regime is identified in regions of drier soil moisture with the CTP-HI space and is indicative of high sensible heat fluxes which result in significant boundary layer growth that can also lead to convection, thus causing negative feedback.

However, this regime is often associated with less frequent and smaller precipitation events, resulting in an overall drying effect. The transitional regime is neither wet nor dry but indicates a neutral L-A coupling regime. Lastly, areas outside the main cluster in CTP-HI space are classified as atmospherically controlled, signifying that the convective processes are predominantly driven by atmospheric factors rather than land surface feedback.

## 2.2 L-A Coupling Strength

One way to define coupling strength in the context of L-A interactions is by the persistency of a specific coupling regime over time. This quantification reflects the intensity and durability of the interactions, where a persistent coupling state indicates a stronger connection between soil moisture levels and atmospheric conditions within the CTP-Hi framework. A high coupling strength signifies a robust, stable interaction that can impact weather patterns and short-term climate variability. In contrast, a low coupling strength indicates weaker and potentially more variable interactions. Examining the persistency of these coupling

states will provide insights into the L-A interactions and their implications for climate modelling and prediction.

The two-dimensional CTP-HI framework (section 2.1) produces distinct land-atmosphere coupling regimes: dry, wet, atmospheric controlled, and transitional. Once the CTP-HI space is classified, a daily time series of the coupling classification is generated using only CTP-HI values based on the location within the classified CTP-HI space for each day. To concentrate the analysis on the dry and wet regimes, the atmospheric controlled and transitional regimes are combined into one regime

called atmospherically controlled. The coupling strength is then quantified by using a three-state (dry, wet, and atmospherically controlled) Markov Chain model. This model operates on the principle that the likelihood of the next day coupling state depends only on the current coupling state and produces transition probabilities between the three states. Of the transitional probabilities, the likelihood of staying in the same state (persistence) provides insights into the strength of the coupling and the L-A feedback for different soil moisture data sets.

## 2.3 Merged CTP and HI


Evaluating the sensitivity of different soil moisture data sets on the coupling strength requires a consistent dataset of CTP-HI. However, the only global CTP and HI datasets are from reanalysis or satellite. Relying solely on a single reanalysis dataset for CTP-HI could introduce bias and limit the comparability when evaluated alongside multiple soil moisture datasets. Moreover, satellite-derived CTP-HI estimates face significant challenges, such as missing observations and lower vertical resolution

(Roundy and Santanello, 2017). These factors can impact the quality and reliability of the data.



To address the limitations inherent in single-source CTP-HI estimates, a merged CTP-HI product using data from three different reanalysis sets is developed. This approach aims to provide a more comprehensive and reliable benchmark for comparing the impact of soil moisture dynamics on L-A coupling. Previous research has shown the Triple Collocation (TC) data merging methodology reliable for combining hydrologic variables without requiring ground-based observational data

(Yilmaz et al., 2012). The TC technique employs a least-squares approach to calculate weight distributions for each dataset based on the root mean square error (RMSE). A core assumption of this method is the independence and lack of correlation between errors across the datasets. This condition is crucial for ensuring the accuracy of the TC method and without it, the merged estimate is prone to biases. Mathematically, the errors within each dataset can be articulated as a linear combination of mutually independent error terms, as mentioned in Equation 1 (Wu et al., 2020).

$$\theta_i = a_i \, b_i \theta + \varepsilon_i \qquad (1)$$

where, $\theta_i$ are collocated measurements of an arbitrary variable (here CTP and HI) for i = 1, 2, and 3 related to the true value $\theta$ with $\varepsilon_i$ as random errors, $a_i$ and, $b_i$ correspond to the intercept and slope obtained through ordinary least squares regression. Considering these assumptions and their potential implications, the resulting merged dataset provides a more comprehensive and accurate representation of the underlying physical phenomenon.

This methodology is employed to create a merged CTP and HI dataset from three reanalysis products: the Modern-Era Retrospective Analysis for Research and Application, version 2 (MERRA2), the Climate Forecast System Reanalysis (CFSR), and European Centre for Medium-Range Weather Forecast (ECMWF) Reanalysis v5 (ERA5). The triple collocation method requires choosing a reference data set (used to estimate the true values of the measured physical phenomenon). MERRA2 was selected to be the reference dataset, however, it is worth noting that previous studies have indicated that the choice of reference

dataset does not impact the outcome of TC error variance (Anderson et al., 2012). The first step involves converting the CFSR and ERA5 estimates (θ) into the MERRA2 climatology (θ') using equations (2) and (3) to establish a common reference framework for the datasets.

$$\theta'_{CFSR} = \mu_{MERRA2} + (\theta_{CFSR} - \mu_{CFSR})\left(\frac{\sigma_{MERRA2}}{\sigma_{CFSR}}\right) \qquad (2)$$

$$\theta'_{ERA5} = \mu_{MERRA2} + (\theta_{ERA5} - \mu_{ERA5})\left(\frac{\sigma_{MERRA2}}{\sigma_{ERA5}}\right) \qquad (3)$$

In the above step, the normalized composites are linearly scaled and used as input in TC analysis, as described below in equations (4) – (6). Each dataset was rescaled to a consistent grid resolution of 1°x1° before applying the TC method and was evaluated during the 2003 to 2022 period to calculate the TC error value ($\varepsilon^2$).

$$\varepsilon^2_{MERRA2} = \{(\theta_{MERRA2} - \theta'_{CFSR})(\theta_{MERRA2} - \theta'_{ERA5})\} \qquad (4)$$

$$\varepsilon^2_{CFSR} = \{(\theta_{CFSR} - \theta'_{MERRA2})(\theta_{CFSR} - \theta'_{ERA5})\} \qquad (5)$$

$$\varepsilon^2_{ERA5} = \{(\theta_{ERA5} - \theta'_{CFSR})(\theta_{ERA5} - \theta'_{MERRA2})\} \qquad (6)$$

The above equation brackets indicate the temporal average of differences between two variables over the study area. Mathematically, the ideal merger of a variable from numerous datasets requires information regarding errors, such that a highly accurate data source receives the larger weight for merging and vice versa (Chen et al., 2022). Besides, to generate the unbiased



merge data product from three datasets, the sum of individual weight at each grid cell should be one ($w_x + w_y + w_z = 1$)

(Gruber et al., 2017). In the context of this analysis, the variables x, y, and z correspond to the datasets MERRA2, CFSR, and ERA5, respectively. The merge outcome or cost function is calculated using equations (7) to (9), which minimizes the error variance in the merging outcome obtained from the least square approach and highly depends on the TC error (Lyu et al., 2021).

$$w_x = \frac{\varepsilon_y^2 \varepsilon_z^2}{\varepsilon_x^2 \varepsilon_y^2 + \varepsilon_x^2 \varepsilon_z^2 + \varepsilon_y^2 \varepsilon_z^2} \tag{7}$$

$$w_y = \frac{\varepsilon_x^2 \varepsilon_z^2}{\varepsilon_x^2 \varepsilon_y^2 + \varepsilon_x^2 \varepsilon_z^2 + \varepsilon_y^2 \varepsilon_z^2} \tag{8}$$

$$w_z = \frac{\varepsilon_x^2 \varepsilon_y^2}{\varepsilon_x^2 \varepsilon_y^2 + \varepsilon_x^2 \varepsilon_z^2 + \varepsilon_y^2 \varepsilon_z^2} \tag{9}$$

The cost function to merge three datasets is,

$$J = \varepsilon^2{}_m = w_x \varepsilon_x^2 + w_y \varepsilon_y^2 + (1 - w_x - w_y) \varepsilon_z^2 \tag{10}$$

The weights remain consistent with the three datasets. Additionally, the study comprises a 30-day centered window (15 days

on either side of the compound event) that removes the effect of seasonality. To reduce complexity due to the leap year and keep consistency, the analysis only considers 28 days in February for each year. The resulting merged CTP and HI, as well as the individual reanalysis products (MERRA2, CFSR, and ERA5) are evaluated against estimates of CTP and HI from in-situ radiosondes and satellite remote sensing based on several summary metrics, including the Mean Absolute Error (MAE), bias, and the correlation coefficient (CC) as discussed in section 4.1.

**3. Dataset Description**

The analysis approach utilizes datasets that include soil moisture products derived from satellite remote sensing and assimilation frameworks, as well as atmospheric profiles crucial for calculating the CTP and HI metrics. The computation of CTP and HI requires surface pressure and vertical profile of humidity (q), and temperature (T), in addition to surface-level data including 2-meter temperature (T2m) and dew point temperature (DP2M). A summary of these datasets, including their

horizontal and temporal resolutions and coverage, is presented in Table 1. To assess the performance of merged CTP-HI the analysis also includes Atmospheric Infrared Sounder Version 7(AIRSv7) satellite remote sensing and radiosonde observations from Integrated Global Radiosonde Archive Version 2 (IGRA2). To ensure consistency in the study, all utilized datasets were standardized to a uniform spatial resolution of 1°x1°, which aligns the analysis with the spatial and temporal coverage of the AIRSv7 from 2003 to 2022.





**Table 1: Description of the dataset for Convective Triggering Potential (CTP), Humidity Index (HI), and Soil Moisture (SM) used in this study.**

| Dataset | Type | Variable | Horizontal Resolution | Vertical Resolution | Temporal Resolution | Temporal range |
|---|---|---|---|---|---|---|
| MERRA2 | Reanalysis | CTP, HI | 0.5°x0.625° | 72 Levels | 6 hours | 2003 to 2022 |
| CFSR | Reanalysis | CTP, HI | 0.5° x 0.5° | 64 Levels | 6 hours | 2003 to 2022 |
| ERA5 | Reanalysis | CTP, HI | 31 km | 137 Levels | 1 hour | 2003 to 2022 |
| IGRA2 | In-situ | CTP, HI | - | N/A | 6-12 hours | 2003 to 2022 |
| AIRSv7 | Remote Sensing | CTP, HI | 1°x1° | 24 Levels | 12 hours | 2003 to 2022 |
| SMAPL3 | Remote Sensing | SM | 9 km | N/A | 12 hours | April 2015 to 2022 |
| SMAPL4 | Assimilated Soil Moisture | SM | 9 km | N/A | 3 hours | April 2015 to 2022 |

### 3.1 CTP and HI Datasets

The CTP and HI is calculated using three reanalysis datasets (MERRA2, CFSR and ERA5), satellite estimates, and in-situ observations. When performing triple collocation analysis, it is crucial to consider the presence of biases and errors in the datasets across different variables and applications. For instance, Park et al. (2020), Dong et al. (2020), Arshad et al. (2021), and Kozubek (2020) have observed variations among these reanalysis datasets in their respective studies. Yingshan et al. (2022) found seasonal trend variations in all three datasets and concluded that ERA5 demonstrated superior performance in short-wave and long-wave radiation compared to MERRA2. Zhang et al. (2021) evaluated the surface air temperature in China and reported significant interannual variability in the MERRA2, CFSR, and ERA5 datasets. Hassler and Lauer (2021) noted that performance in tropical areas varies depending on the subset of data used, such as land-only, ocean-only, or land-atmosphere-ocean. Santanello et al. (2015) reported a dry bias in CFSR and a wet bias in MERRA in the overall surface planetary boundary layer (PBL) based on local land-atmosphere coupling (LoCo) analyses over the U.S. Southern Great Plains. A description of each of the CTP-Hi datasets is given below.

### 3.1.1 The Modern-Era Retrospective Analysis for Research and Application, version 2 (MERRA2)

NASA's Global Modelling and Assimilation Office (GMAO) developed MERRA2 as an atmospheric reanalysis dataset, employing the Goddard Earth Observing System (GEOS) Catchment Land Surface model. MERRA2 provides 6-hourly observations with an approximate spatial resolution of 0.5°x0.625° and includes 72 hybrid pressure levels ranging from the surface to 0.01hPa (Ochege et al., 2021). The data assimilation system of MERRA2 utilizes the 3D-var algorithm and spans from 1980 to the present. Gelaro et al. (2017) describe the assimilation process of MERRA2, along with advancements and improvements made in the system.





### 3.1.2 The Climate Forecast System Reanalysis (CFSR)

The Climate Forecast System Reanalysis (CFSR) is developed by the National Center for Environmental Prediction (NCEP). It covers the period from 1979 to the present. It provides six-hourly variables estimations, including 64 atmospheric levels at a 0.5° x 0.5° horizontal resolution (Centella-Artola et al., 2020). Operating as a global coupled atmosphere-ocean-land surface-
sea ice system, CFSR incorporates satellite radiance data and employs the Integrated Forecasting System (IFS) Cycle 41r2 with the 3D-var data assimilation system. Observations are carefully considered for each component during the assimilation process of the CFSR dataset, as highlighted in Saha et al. (2010). CFSR is a homogeneous reanalysis product suited for long-term climate studies, as observations play a crucial role in the assimilation process.

### 3.1.3 European Centre for Medium-Range Weather Forecast (ECMWF) Reanalysis v5 (ERA5)

ERA5, the fifth ECMWF reanalysis data of global climate, is accessible from January 1959 to the present and produced by the Copernicus Climate Change Service (C3S). ERA5 provides hourly land and atmospheric climate variable estimations at approximately a 31 km spatial resolution and 137 levels from the surface to 80 km (Centella-Artola et al., 2020). It employs the Integrated Forecasting System (IFS) Cycle 41r2 and assimilates satellite and in-situ observations. ERA5 includes advanced screen-level assimilation for 2m temperature and relative humidity components, where the soil moisture is nudged to better
match the 2-meter observations. Hersbach et al. (2020) compared ERA5 with radiosonde data and showed temperature, wind, and humidity improvements in the troposphere for the latest version.

### 3.1.4 The Integrated Global Radiosonde Archive (IGRA) Version 2

The Integrated Global Radiosonde Archive Version 2 (IGRA2) is a comprehensive dataset provided by the National Center for Environmental Information (NCEI) of the National Oceanic and Atmospheric Administration (NOAA) in the United States.
It offers access to high-quality radiosonde observations from over 1500 stations worldwide from 1905 to the present (Durre and Yin, 2008). The dataset has undergone quality control and adjustments to correct instrument biases across various regions. These procedures ensure that the IGRA records are homogeneous and robust, making them valuable for long-term climate studies.

Despite its comprehensive coverage, the IGRA2 dataset presents challenges, including non-uniform data distribution across
the Globe due to variable observation frequencies. Radiosonde observations are typically recorded twice daily at 00:00 and 12:00 Coordinated Universal Time (UTC); in some regions, additional observations are taken at 06:00 and 18:00 UTC. Therefore, the frequency and timing of these observations vary among stations and locations. Calculations for CTP and HI are performed using radiosonde data that fall within a ±3-hour window of the night-time AIRS overpass (~0130 UTC). This targeted approach aligns CTP and HI calculations with the same observation time during the merging and validation process.
Also, when multiple observations are available for a single grid cell, the selection criterion prioritizes the station offering the most frequent data. A map of the geographic location of the IGRA2 radiosonde observation sites across the Globe, along with





regional totals in North America (NAM), South America (SAM), Africa (AFR), Europe (EUR), Asia (ASA), and Australia (AUS) is given in Fig. 2. Across the world, 534 locations have data available from 2003 to 2022. Most observations are in Asia, followed by North America and Europe.




**Figure 2: IGRA2 observation stations across the Globe, along with regions selected for further analysis and the number of observation stations in each region.**

### 3.1.5 Atmospheric Infrared Sounder Version 7 (AIRSv7)

AIRS was launched in 2002 on NASA's Aqua satellite. Working with AMSU-A, AIRS retrieved thermodynamic profiles (temperature and humidity) using passive radiance observations. This study focuses on the descending orbit, covering Northern to Southern Hemisphere movement with an equator crossing at 1:30 am. Data has a 2x/daily temporal resolution, capturing half of the 8-day Aqua orbit repeat cycle. Level 3 files contain averaged quality and geophysical parameters in 1°x1° grid cells, including humidity, geopotential height, and temperature profiles at 24 pressure levels from 1 to 1000 hPa (AIRS Project, 270 2020). In Version 7, short-wave exclusion in the retrieval algorithm reduces bias, and targeted channel selection focused on water vapor retrieval improves temperature/water vapor profile performance (Zhang et al., 2023a). It should be noted that for the processing and analysis of the AIRSv7 data, no alterations were made to the predefined quality control (QC) flags.

### 3.2 Soil Moisture Active Passive (SMAP)

NASA's Soil Moisture Active Passive (SMAP) mission provides global monitoring mapping of soil moisture content. The 275 Enhanced SMAP Level 3 (SMAPL3) products, derived from the foundational Level 1 and 2 data, provide standardized, gridded global soil moisture (O'Neill et al. 2021) at 9 km resolution with the capability to observe the global Earth surface every 2-3 days. SMAP Level 4 (SMAPL4) data are derived from a land surface modelling system that assimilates brightness temperatures. SMAPL4 integrates SMAP Level 2 brightness temperature measurements, along with initialization and forcing inputs for the Catchment Land Surface Model (CLSM) (Reichle et al. 2017), thus producing 3-hourly, comprehensive estimates





of surface and root-zone soil moisture at a 9 km spatial resolution (Qiu et al., 2021). SMAPL3 provides direct satellite remote
       sensing retrievals of surface soil moisture. At the same time, SMAPL4 data builds upon this with a sophisticated data
       assimilation system. This differentiation is vital for our analysis, enabling us to assess the benefits of incorporating the
       observations into the modelling system in capturing soil moisture dynamics and the resulting impact on L-A coupling.

       It is also essential to recognize that SMAPL3 has missing observations due to the satellite's orbital constraints, which is not

the case with the comprehensive SMAPL4 dataset spanning from April 2015 to 2022. This variance in data completeness could
       affect the comparative analysis. To address this, we introduce a synthetic data set labelled SMAPL4_L3, which utilizes
       SMAPL4 but with the temporal coverage of SMAPL3 data, ensuring a comparison with an equal sample of soil moisture
       estimates. This study uses the morning observations, specifically from the sunrise overpass, of SMAP soil moisture data, which
       are crucial in capturing its interactions with the atmospheric processes. The different SMAP data products are used to classify

the CTP-HI framework from 2015 to 2022 for all datasets to assure consistency with the analysis. Since soil moisture
       measurements are only needed for the classification period, the time series of coupling classification can still cover the entire
       time series of CTP-HI from 2003 to 2022.

## 4. Results

### 4.1 Evaluation of Merged CTP-HI

The foundational concept of the TC method emphasizes that inaccuracies in individual datasets should remain uncorrelated to
       avoid the amplification of errors in the aggregated outcome from multiple sources. The merged dataset, derived from three
       reanalysis datasets, will likely exhibit correlated errors. These errors were analysed across the datasets MERRA2, CFSR, and
       ERA5 at the AIRS overpass time (~1:30 AM local time), using observational data from IGRA2 at a location in Kansas, United
       States (coordinates 39.96, -95.26, referred to in Fig. 3(a). The analysis indicates a significant correlation of errors, with a

correlation coefficient of around 0.85 when assessed against IGRA2 data. This correlation appears more pronounced for the
       CTP than the HI, suggesting a stronger association of errors within the stability metric (CTP). This may stem from similar
       errors in assimilating temperature and humidity profiles or their radiances, while HI may be influenced more by differences in
       model physics and parameters, screen-level nudging (in ERA5), compounded by challenges in assimilating near-surface
       humidity data.

Fig. 3(b) displays observed error correlation across the Globe (534 IGRA2 locations), with most locations having a correlation
       exceeding 0.7. This high correlation points to similar error sources in the datasets for CTP and HI across the three datasets
       considered.  The reason for the similarities across the different reanalysis lies in their assimilation of the same sources of
       satellite data. Despite employing diverse methods of Data Assimilation (DA) and featuring distinct physics, parameters, etc.,
       the common input of satellite data contributes to this consistency. Despite these correlated errors, the triple collocation method

helps integrate the three reanalysis datasets. However, it is essential to acknowledge that the correlated nature of the errors will
       affect the accuracy of the resulting merged dataset.

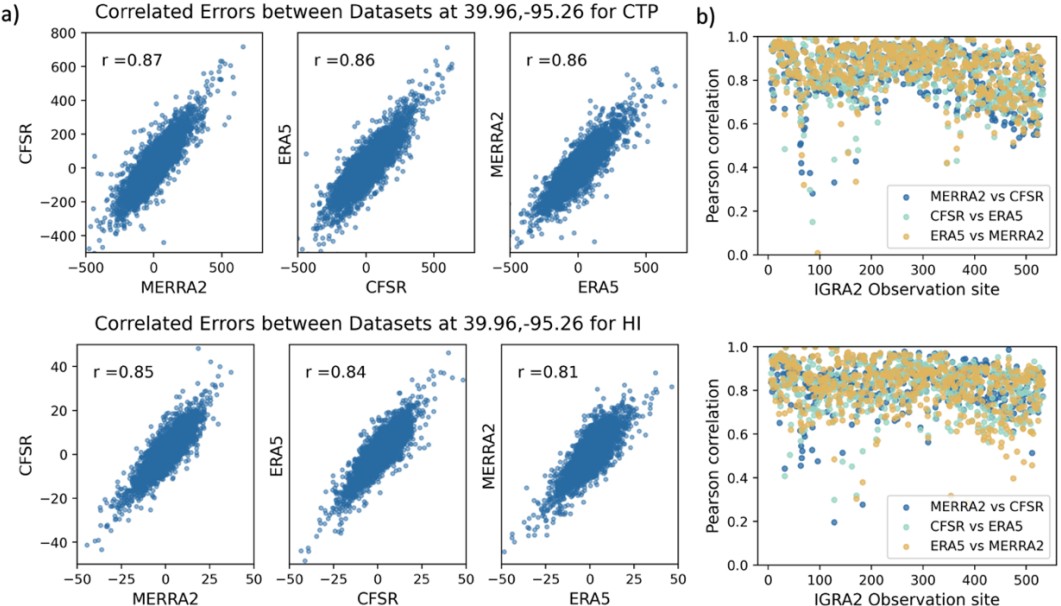

**Figure 3: (a) Observed errors between IGRA2 and reanalysis datasets at a location in Kansas, United States (Latitude: 39.96, Longitude: -95.26) for CTP and HI from 2003 to 2022, along with the Pearson correlation displayed in the upper left corner. (b) Observed error correlation between reanalysis data sets at IGRA2 stations across the Globe (529 sites).**

The triple collocation-based error (TC error) using equations (4)-(6) given in the supplementary material (Fig. S1) reveals a remarkably high TC error for the CTP within the MERRA2 and CFSR datasets, most notably over the northern hemisphere and South America. For the merged product construction, grid cells are weighted according to the error variance calculated by equations (7)-(9). Those with a higher error variance are assigned lower weights, reflecting their reduced reliability. Conversely, grid cells with lower error variance are deemed more reliable and thus are given greater weights in the merging process.

Table 2 demonstrates the distribution of TC-based weights for CTP and HI across various global regions for the three datasets. The spatial maps in the supplementary material (Fig. S2) details the weights for each reanalysis product for CTP and HI globally. These weights are directly proportional to the relative error variance; areas where MERRA2 and CFSR show larger variances (Fig. S3), especially in the northern hemisphere and South America, tend to favour ERA5 regarding weight for the CTP. Based on Table 2, ERA5 emerges as the leading dataset for CTP, being allocated the highest weight in most regions. However, in Europe and Africa, the weight distribution for CTP is almost similar among the three datasets, indicating a balanced reliance on each dataset within these continents.





**Table 2: Average weight across North America (NAM), South America (SAM), Africa (AFR), Europe (EUR), Asia (ASA), and Australia (AUS). The color gradient is applied based on the minimum value (yellow color) to the maximum value (green color) for CTP and HI for AIRS overpass.**

| Region | NAM | SAM | AFR | EUR | ASA | AUS |
|--------|------|------|------|------|------|------|
| Weight distribution across the region for CTP | | | | | | |
| MERRA2 | 0.2724 | 0.2745 | 0.3241 | 0.3528 | 0.2816 | 0.3111 |
| CFSR | 0.2964 | 0.2992 | 0.3241 | 0.3372 | 0.3019 | 0.3111 |
| ERA5 | 0.4311 | 0.4262 | 0.3496 | 0.3099 | 0.4164 | 0.4003 |
| Weight distribution across the region for HI | | | | | | |
| Region | NAM | SAM | AFR | EUR | ASA | AUS |
| MERRA2 | 0.3289 | 0.2883 | 0.3471 | 0.3663 | 0.354 | 0.3054 |
| CFSR | 0.3125 | 0.3124 | 0.3318 | 0.3387 | 0.3005 | 0.3378 |
| ERA5 | 0.3584 | 0.3991 | 0.321 | 0.2948 | 0.3453 | 0.3567 |

The weight allocation for the HI shows considerable regional variation. In South America, the MERRA2 dataset is assigned the lowest weight, implying that it contributes less to the combined HI product. Conversely, in Europe, it is the ERA5 dataset that receives the lowest weight, signifying its reduced contribution to the HI variable in this region. These regional differences in the weighting of datasets underscore the merging process, allowing for a location-specific approach to creating a merged product.

### 4.1.2 Performance of Merged CTP-HI

The data is merged following equation (10), and the resultant spatial distribution of average CTP and HI during the summer season (June, July, and August of JJA) for the year 2012 is portrayed in supplementary material (Fig S3 and S4). This figure compares the CTP and HI values derived from the MERRA2, CFSR, and ERA5 datasets alongside the merged product.

The accuracy of the merged dataset, alongside the individual reanalysis datasets MERRA2, CFSR, and ERA5, is evaluated through a comparison with IGRA2 radiosonde and AIRSv7 satellite observations, as depicted in Fig. 4 and 5. The evaluation utilizes performance metrics such as the Mean Absolute Error (MAE), bias, and Correlation Coefficient (CC) across major global regions. The results indicate that the merged dataset consistently achieves the lowest MAE for CTP and HI variables, outperforming the individual reanalysis datasets that perform best independently. This improvement in accuracy suggests that the merging process effectively consolidates the strengths of the individual datasets while mitigating their respective biases despite the correlated errors seen in Fig. 3.

The merged product compensates for these inaccuracies in regions where the MERRA2 dataset exhibits substantial discrepancies, demonstrating a practical integration approach that improves the overall metric representation. Furthermore, the merged dataset shows a considerable reduction in positive bias for CTP across regions like South America, Europe, and Asia, as well as a decrease in negative bias for HI in most regions, except for Asia. These improvements highlight the efficacy of





the merging process in yielding a more reliable dataset, which is especially beneficial for L-A coupling studies in regions challenged by less accurate reanalysis data.

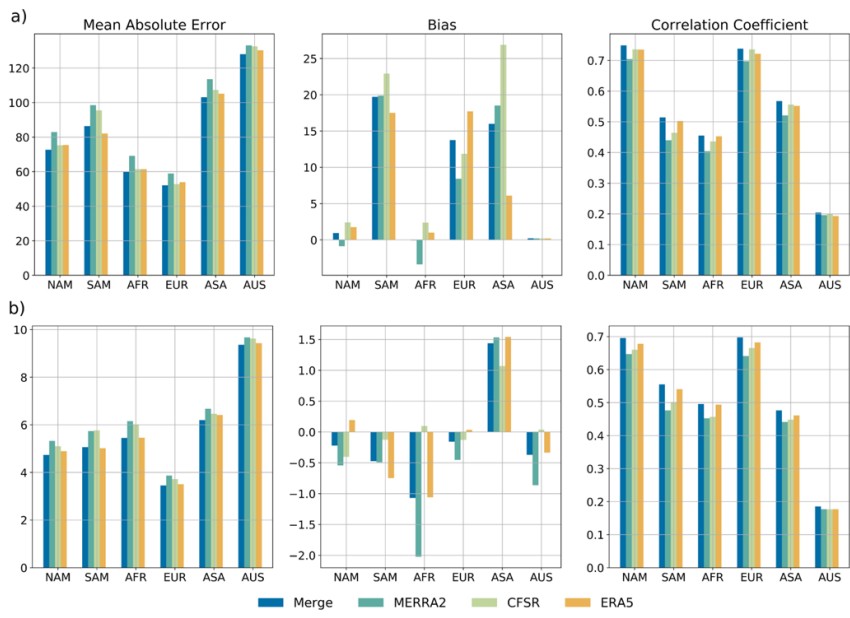


**Figure 4: Bar plot of performance metric and intercomparison of merged data and reanalysis with radiosonde observation from IGRA2 in each global region for (a) CTP and (b) HI.**

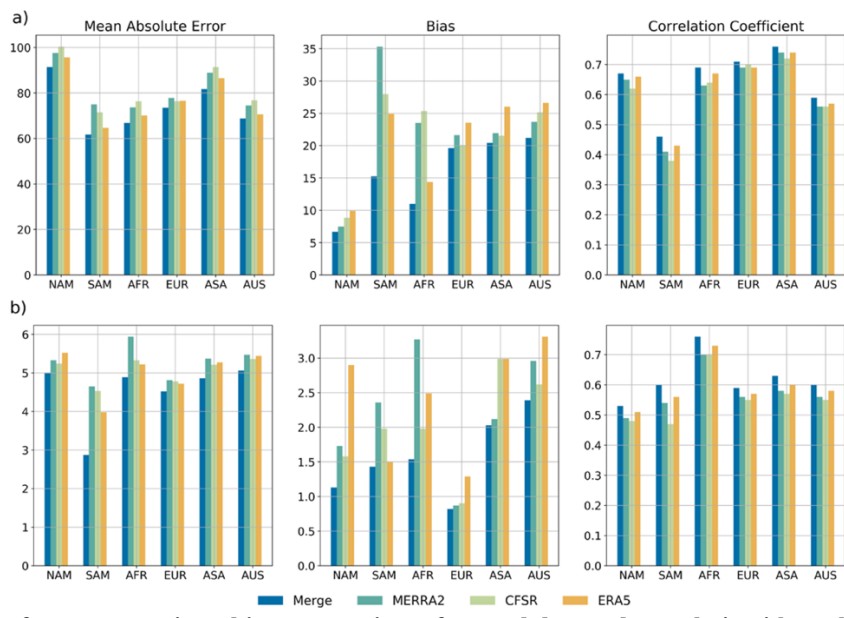

**Figure 5: Bar plot of performance metric and intercomparison of merged data and reanalysis with satellite remote sensing from**
**AIRSv7 in each global region for (a) CTP and (b) HI.**





Fig. 5 presents the same set of accuracy metrics as in Fig. 4 but compared against the AIRSv7 data. The figure indicates a general trend of overestimation for CTP and HI by the reanalysis and merged datasets across all regions as compared to AIRSv7. Acknowledging the limitations in measurement capabilities and inherent biases of AIRSv7, particularly in terms of lower troposphere retrieval, which may differ from those in the other datasets, is essential. For example, the limited sensitivity
and vertical resolution of AIRSv7 in the PBL might lead to positive biases, resulting in discrepancies with the reanalysis datasets vs. that observed with radiosonde data as the reference.

Regarding temporal correlation, the merged dataset and the ERA5 reanalysis score the highest values, reflecting a consistent signal with observations from IGRA2 and AIRSv7 (as shown in Fig. 4 and 5. When evaluated against both ground-based observations and satellite remote sensing, there is a noticeable improvement in the performance of the merged product
compared to the individual reanalysis datasets. The merged product demonstrates a closer alignment with the ground truth, as represented by IGRA2, capturing the observed signal with greater accuracy. This enhancement in accuracy emphasizes the merged product's value as a reliable and comprehensive dataset, which is especially beneficial for future research on L-A coupling.

## 4.2 Coupling Strength in the Contiguous United States

The previous section evaluated the merged CTP and HI products at the AIRS overpass time (~1:30AM), however, the theory of the CTP-HI framework for classifying coupling relies on early morning observations of the CTP and HI (Findell and Eltair, 2003) which also aligns with the SMAP overpass time (~6AM local time). Table S5 gives the weights for the merged product at sunrise. As compared to the AIRS overpass weights (Fig. 4), the sunrise weights show similar spatial patterns with marginal variations. This is consistent with previous research by Roundy and Santanello (2017) that demonstrated some differences
between the AIRS and sunrise overpasses estimates of CTP and HI, yet acknowledged the overarching similarity in their patterns. Thus, the remainder of the analysis will use a merged product of sunrise CTP and HI to classify the L-A coupling.

The merged CTP and HI are combined with three different datasets of soil moisture (SMAPL3, SMAPL4_L3, and SMAPL4) to create a timeseries of L-A coupling classification from 2003-2022. Fig. 6 examines the L-A coupling strength of the timeseries based on the persistence of the dry and wet regimes over the contiguous United States for the different soil moisture
datasets. Fig. 6(a) reveals a consistent pattern of persistent dry conditions in the inter-mountain west region and persistent wet conditions in the north-western and eastern parts of the country. A side-by-side evaluation shows that the SMAPL4 dataset displays a more persistent pattern under dry and wet conditions. This indicates an enhanced representation of L-A coupling in SMAPL4 that may be due to the strong vertical coupling of soil moisture in the Catchment assimilation processes. On the other hand, SMAPL4_L3 data shows less persistence than SMAPL4, suggesting that part of observed difference between SMAPL3
and SMAPL4 is due to sample size. Despite these differences, the overarching spatial pattern remains consistent across all datasets evaluated.

To delve deeper into the observed differences in coupling strength, the lagged correlation between soil moisture and the CTP and HI for the three data sets is shown in Fig. 6(b) and 6(c). Because reliable weather predictability is generally limited to 10





days (Krishnamurthy, 2019), the 10-day average lag correlation over the contiguous US is analysed to identify variable-
dependent predictability of CTP and HI from soil moisture (positive lag) and the predictability of soil moisture informed by
CTP and HI (negative lag). For both the CTP and HI, the correlation with soil moisture is negative due to the relationship
between SM-CTP and SM-HI. Wet soil typically results in surface cooling when solar radiation is limited, leading to a more
stable temperature profile in the lower atmosphere. This stability restricts vertical movement and consequently leads to a lower
CTP, thus creating a negative correlation. HI, on the other hand, measures atmospheric moisture content. Higher HI values
signify drier air, while lower values indicate moisture-rich air closer to saturation. High soil moisture enhances evaporation,
which adds water vapor to the atmosphere, reducing the gap between temperature and dew point and thus lowering the HI,
resulting in a negative correlation. The results in Fig. 6(b) and 6(c) suggest that soil moisture is a more reliable predictor of
CTP and HI, as shown by the larger magnitudes of lag correlations. Among the datasets examined, SMAPL4 consistently
shows higher correlations at all lag intervals, further suggesting that the assimilation process yields a stronger relationship in
the temporal dynamics between the land surface and the atmosphere. In contrast, this pattern of a stronger L-A connection is
less evident with SMAPL3, especially for the CTP.

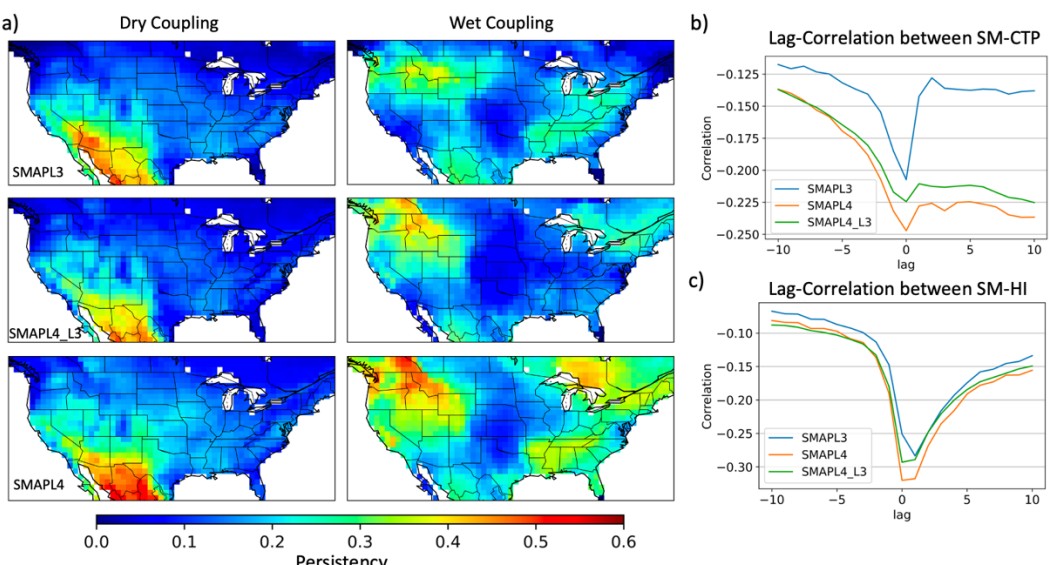

**Figure 6: Comparative analysis of Land-Atmosphere coupling strength using SMAPL3, SMAPL4_L3, and SMAPL4 data (a)**
**persistence probability for the dry and wet coupling regime and (b and c) lag- correlation between soil moisture and CTP and HI**
**across the contiguous United States.**

Fig. 7 delves into a grid-scale analysis at the specific coordinates (38.89, -115.59), located in Nevada, USA, which shows
notable differences in the dry and wet coupling regimes. Fig. 7(a) shows the classified CTP-HI space based on SMAPL3,
SMAPL4_L3, and SMAPL4 and clearly distinguishes the coupling regimes in the CTP-HI space across the datasets. Not
surprisingly, SMAPL4 indicates more bins assigned to wet and dry coupling regimes, thereby indicating an increased coupling





strength within the observed time series. Fig. 7(b) illustrates the relationship between soil moisture and the CTP-HI space. States of wet coupling are associated with higher soil moisture levels in the combined probability space of CTP-HI-SM, whereas dry states are linked to lower moisture levels. The disparities in the distribution of soil moisture are quite evident in Fig. 7(c). The SMAPL3 dataset shows a tendency for observations to skew toward the lower end of the soil moisture spectrum.

At the same time, SMAPL4 tends to exhibit a clustering of observations in the mid-range, between 0.4 to 0.8. There is also very little difference in soil moisture distribution due to sample size (Fig. 7(c)), but more of a difference within the CTP-HI space (Fig. 7(b)).

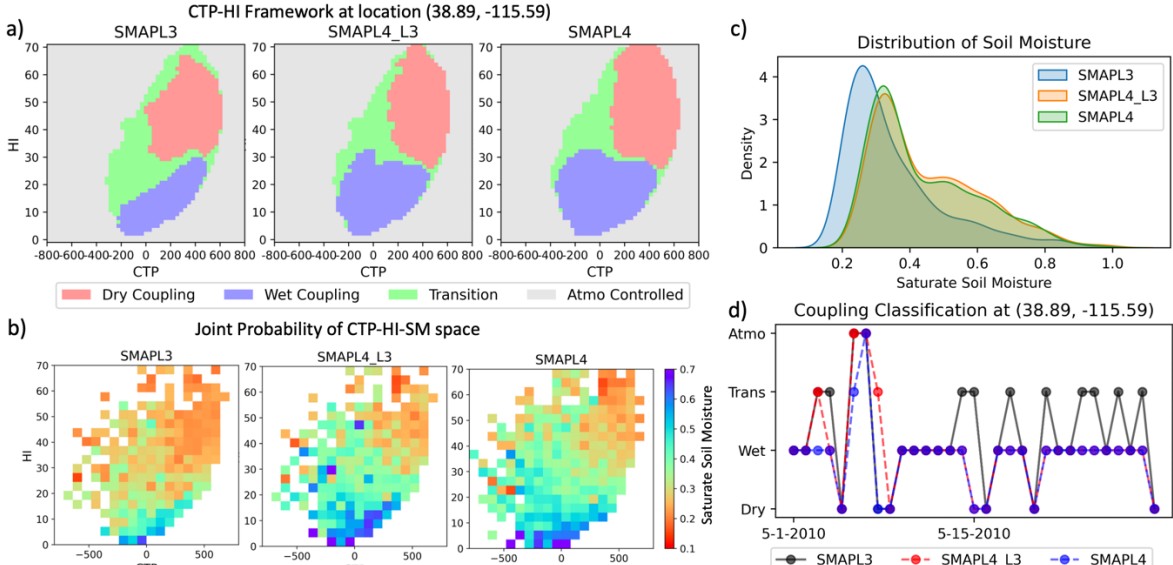

**Figure 7: Multifaceted assessment of coupling classification influenced by soil moisture across SMAPL3, SMAPL4_L3, and SMAPL4 data at a specific grid location (38.89, -115.59) situated in Nevada, USA (a) CTP-HI framework (b) A joint probability space of CTP-HI-SM (c) Saturate soil moisture distribution (d) Coupling classification time series for May 2010.**

These differences in both the soil moisture distribution and its projection in the CTP-HI space affect the classification of coupling regimes and therefore the coupling strength of the timeseries. Fig. 7(d) depicts the daily coupling classification for 430 an arbitrary month (May 2010). The SMAPL4 dataset, with its higher number of wet regime classifications, demonstrates a greater likelihood of days being categorized as wet. This is evidenced in the time series, where most days are classified under wet conditions in SMAPL4, in contrast to the SMAPL3 dataset, which indicates more days in a transition state. Sample size also only has a small impact on the classification, with only three days being different between SMAPL4_L3 and SMAPL4. This variance underscores the importance of soil moisture measurements on the daily classification of L-A coupling within the 435 CTP-HI framework.





### 4.3 Global Coupling Strength

The previous section highlighted the connection between coupling strength (persistence) and the lag-correlation between soil moisture and the CTP and HI. To explore this further, Fig. 8 presents the average coupling strength (for both dry and wet regimes) on the x-axis and the average positive (soil moisture predicts CTP-HI) lag correlation on the y-axis. Overall, all soil

moisture datasets show a non-linear relationship between coupling strength and lag correlation, with little relationship between the variables for low coupling strength that transitions into a stronger relationship as coupling strength increases. To help quantify this, the data is fit to a quadratic model, and the $R^2$ for this relationship is shown. The $R^2$ value is calculated to measure how well the variance in the observed data can be explained by the quadratic model.  For all soil moisture datasets, the soil moisture CTP relationship is weak as indicated by lower $R^2$ (explained variance by the regression line) and the shallow slope

of the regression line. In contrast, the soil moisture-HI relationship is stronger, with higher $R^2$ and more pronounced nonlinear relationship. This indicates that persistency as a measure of coupling strength is predominantly driven by the soil moisture-HI relationship, suggesting a direct influence of soil moisture on lower-level humidity and its correlation over time, as opposed to the more complex influences on atmospheric stability (CTP), which may be more influenced by larger-scale atmospheric conditions. Furthermore, SMAPL4 shows a stronger relationship as compared to SMAPL3, that is slightly impacted by the

difference in sample size between SMAPL3 and SMAPL4.

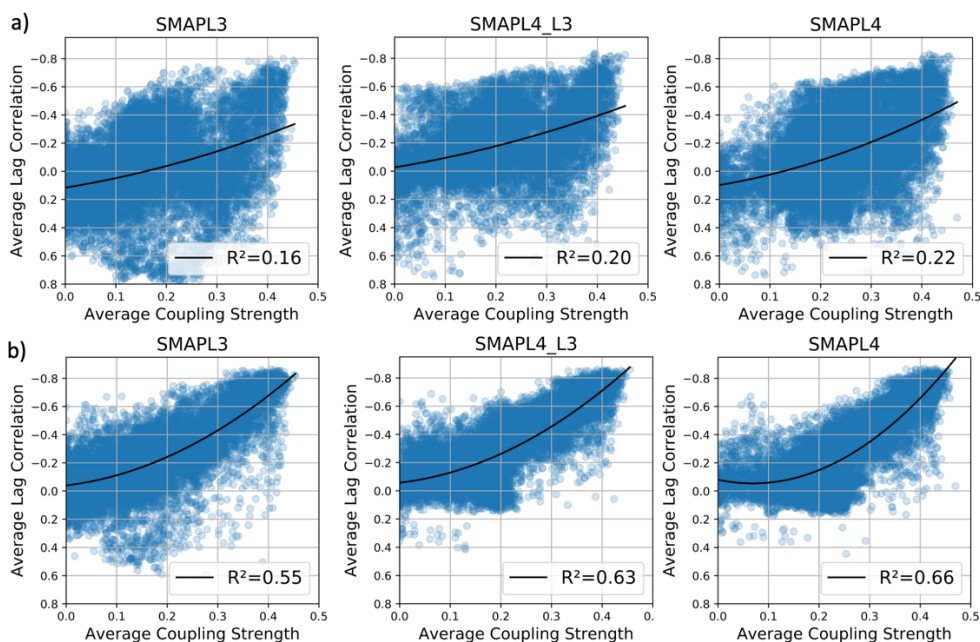

**Figure 8: Comparative analysis of average dry and wet coupling strength and 10-day lag correlation using SMAPL3, SMAPL4_L3, and SMAPL4 soil moisture datasets for (a) CTP and (b) HI.**



Fig. 9 shows the average coupling strength for dry and wet regimes across the global regions from SMAPL3, SMAPL4_L3, and SMAPL4. All datasets are consistent in showing that Africa has the largest average coupling strength, while North America has the smallest average coupling strength. Yet, there are differences in the relative strength for other regions. Notably, the SMAPL4 dataset demonstrates a stronger coupling strength in both the dry and wet regimes, indicating a stronger temporal persistence in the coupling regime. However, when considering the impact of sample size, the difference in coupling strength

is dimensioned. This is particularly true for Africa, where there is little difference between the average coupling strength between SMAPL3 and SMAPL4_L3. In contrast, North America shows the largest difference in coupling strength between SMAPL3 and SMAPL4_L3. As with other regions of the globe, there is little difference in the coupling strength for the dry regime but is predominantly seen in the wet regime coupling strength. This is consistent with Fig. 6, which shows the spatial variability of coupling strength over the contiguous United States.


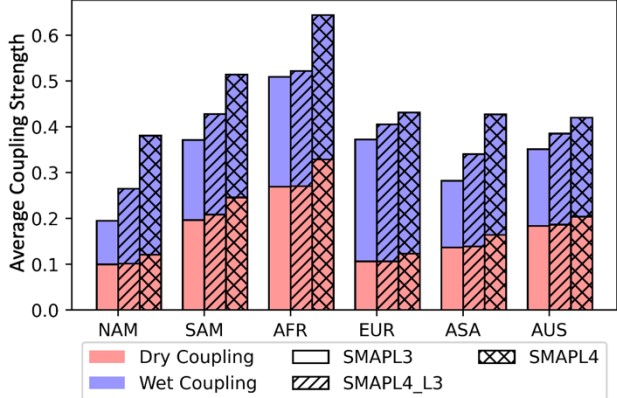

**Figure 9: Comparative analysis of Land-Atmosphere average coupling strength using different soil moisture from SMAPL3, SMAPL4_L3, and SMAPL4 data over the various regions across the Globe.**

**Discussion**

The goal of this analysis is to explore the role of soil moisture from SMAP in quantifying the L-A coupling strength across the globe. As depicted in Figs. 6 through 9, our findings underscore the pivotal role of soil moisture in the representation of L-A coupling strength within the CTP-HI framework developed by Roundy et al. (2013). The coupling strength, as quantified in this study by the persistence in the regime, was shown be related to ability of soil moisture to predict future CTP and HI out to 10 days (Figs. 7 and 8). The observed lag correlation is most pronounced for the soil moisture-HI relationship, indicating

the direct and consistent influence of soil moisture on lower-level humidity across time. In contrast to the more variable effects on atmospheric stability as indicated by the CTP, this suggests that soil moisture has a more straightforward impact on available moisture in the lower atmosphere. Reinforcing this point, Entekhabi et al. (1996) discuss the significant feedback mechanisms between soil moisture and atmospheric processes, highlighting how changes in soil moisture can directly impact the atmospheric environment. This relationship underscores the critical role of soil moisture in influencing atmospheric conditions,





especially humidity levels within the boundary layer. The findings of Wang et al. (2024) further substantiate this argument, which demonstrates the potential for soil moisture to predict the future coupling state and the utility of the persistence regime as a measure of coupling strength.

The research conducted by Dong and Crow, (2019) establishes a foundational understanding of the significance of high signal-to-noise ratios in L-band SMAP soil moisture data. They emphasize that such precision is critical for enhancing the ability to

accurately quantify L-A coupling strengths. This finding underpins the broader investigation into the SMAP products, particularly the differences in response between SMAPL3 and SMAPL4 datasets with respect to L-A coupling strength.

Furthermore, the SMAPL4 dataset exhibits higher coupling strength in both the wet and dry regime across the main global regions (Fig. 9). This indicates that the SMAPL4 has a stronger L-A coupling with the merged CTP-HI product and a greater potential to provide short-term prediction of future coupling states as compared to the SMAPL3 product. The higher coupling

strength observed during the dry coupling regimes in SMAPL4 compared to SMAPL3 can be attributed to the increased temporal frequency used in the analysis. The increased temporal frequency, due to the assimilation process, allows for a more detailed representation of soil moisture dynamics and its interaction with the atmosphere, facilitating a more complete quantification of the L-A coupling processes. This conclusion align with the insight provided by Findell et al.(2023), who emphasized the importance of high-frequency data output for accurately assessing land-atmosphere coupling in climate

models. In contrast, the higher coupling strength observed in the wet coupling regime in SMAPL4 compared to SMAPL3, extends beyond just the contributions from higher data frequency, suggesting other factors at play that require further investigation to fully understand their impact on L-A coupling strength.

However, it is important to consider that the representation of strong coupling might not always accurately mirror the complexities of real-world environmental conditions (Van Vuuren et al., 2012). The accuracy of the stronger coupling in

SMAPL4 is difficult to quantity due to the scarcity of in-situ observations across the globe where simultaneous atmospheric profile observations and soil moisture measurements can be obtained. This limitation of comprehensive ground-based observations poses a significant challenge in validating the true representation of coupling and understanding the intricate interplay between soil moisture and atmospheric conditions (Santanello et al. 2018; Beamesderfer et al. 2022). Consequently, our ability to ascertain which dataset offers a more accurate representation of L-A coupling remains a subject of ongoing

investigation.

In synthesizing the comparison between SMAPL3 and SMAPL4, as depicted in Fig. 7c, highlights the differences in soil moisture representation arise mainly from their distinct constraints and processing methodologies. SMAPL3 tends to skew towards drier values, likely due to the sensor's focus on the topsoil layer, which tends to dry quickly after rainfall events. This skew is influenced by fixed conditions in the retrieval process such as the prescribed freeze/thaw condition and water phase in

the retrieval process, as well as land surface characteristics like vegetation cover and soil properties used in the algorithm. Conversely, SMAPL4 offers a more detailed representation of soil moisture dynamics by incorporating model-based soil hydraulic parameters. These enhancements allow SMAPL4 to more accurately depict variations in soil moisture across different landscapes. Notably, studies by Reichle et al. (2017), Reichle et al. (2019), Reichle et al. (2021) have effectively





reduced bias and expanded the dynamic range of surface soil moisture estimates. While acknowledging the presence of
uncertainties due to soil hydraulic parameter (Loosvelt et al., 2011), these advancements signify considerable progress in refining the precision of soil moisture.

Recent research underscores the vital role of the SMAPL3 soil moisture product in agricultural applications, as demonstrated by Zhu et al. (2024). The accuracy of soil moisture measurement is crucial, and Tavakol et al. (2019) highlighted the SMAPL3 and SMAPL4 soil moisture products are at the forefront of soil moisture accuracy. For instance, Xu (2020) has shown that
SMAPL4's bias is significantly reduced, exhibiting a 46% decrease in surface soil moisture uncertainty. This enhanced accuracy has been corroborated by Zhang et al. (2017), which showed that SMAPL4 captures spatial and temporal soil moisture variations more reliably across the United States. Moreover, SMAP outperforms other soil moisture products, such as AMSR2 L3 and SMOS-IC, across most of the global land surface (Kim et al., 2021). Importantly, SMAP observation could improve drought monitoring and provide detailed insight of drought conditions, as demonstrated in studies by Mishra et al. (2017),
Velpuri et al. (2016), Mladenova et al.(2020). The SMAP observations can crucially enhance the representation of L-A interaction, providing deeper insights into how land surface conditions influence atmospheric responses (Zhang et al., 2023b). The SMAP provides enhanced depiction of L-A coupling through dynamic soil moisture data, offering improved drought monitoring and weather prediction. However, it's crucial to recognize SMAPL4's potential overestimation of L-A coupling strength due to its dependence on model parameterizations, as compared to SMAPL3. This nuanced understanding is crucial
for leveraging SMAP's benefits in practical applications effectively.

Another important aspect of this analysis is the creation of the merged CTP and HI product. The merging process employs the triple collocation method based on the relative errors among the MERRA2, CFSR, and ERA5 datasets. The variations in the weight distributions for CTP and the HI reflect the inherent differences in the datasets. In the merged dataset, the ERA5 reanalysis had the higher weight which may be linked to improved representations of tropospheric temperature and humidity
(Hersbach et al. 2020). Nonetheless, it's critical to understand that a higher weight for ERA5 does not mean that the merged product will closely resemble ERA5 in its characteristics. The merging process involves integrating information from multiple datasets, and the resulting merge product behaves as a distinct and independent entity. Fig. 3 also reveals the presence of correlated errors when compared with radiosonde observations, suggesting the potential for a biased merged product. However, the merged product outperformed the individual reanalysis data sets compared to radiosondes and satellite-based estimates of
CTP and HI (Figs 4 and 5). This suggests that the merging process diminishes the bias arising from the different reanalysis products despite correlated errors and provides a more accurate product. While this validation of the merged product is robust, it is not without its flaws. The differences in spatial and temporal resolutions between the merged product and the IGRA2 radiosonde and AIRSv7 observations is prone to uncertainty. Despite this, the merged dataset still demonstrates a more accurate reflection of in-situ and satellite observations of CTP and HI, thus providing a temporal and spatially consistent dataset for
analysing L-A coupling.





## Conclusion

The development of a merged reanalysis-based product for CTP and HI, which surpasses the performance of individual reanalysis products as evidenced by validations against radiosonde and satellite observations, underscores the potential for TC-based merging approach in global studies. This is shown by integrating the merged CTP and HI metric with SMAP soil moisture datasets for analysing the coupling strength over the globe. The results indicate that SMAPL4 demonstrates a notable higher lag-correlation between soil moisture and the CTP-HI metrics, contributing to the persistence coupling behaviour. This suggests that temporal consistency plays a crucial role in L-A coupling strength. The coupling strength metric is also shown to be sensitive to the number of observations. Despite analysing similar samples from both SMAP product, SMAPL4 consistently presents a stronger representation of coupling strength. Such stronger persistence of wet and dry coupling regimes, as observed in SMAPL4 is not only a result of a greater number of observations, but it possibility due to the distinctive assimilation techniques employed in the SMAPL4 dataset. These insights underscore the impact of soil moisture data on the classification of L-A coupling and its subsequent influence on coupling strength using the CTP-HI framework.

Nevertheless, this research has also shed light on the constraints imposed by the lack of comprehensive ground-based observations. This limitation restricts the capability to fully validate the L-A interactions represented by reanalysis and satellite remote sensing datasets on a global scale. The present work, therefore, underscores the necessity for enhanced observation networks to strengthen the validation processes.

Looking ahead, more research is needed into how the difference and similarities in the SMAP coupling strength will influence the ability to monitor the initiation, intensification, and abatement of drought conditions using the L-A coupling based drought index (Roundy et al., 2013). This drought index is anticipated to be instrumental in deciphering the mechanisms of drought development and how soil moisture dynamics can influence the ability to quantify the severity and duration of droughts. Moreover, a comparative analysis between drought forecasts derived from SMAPL3 and SMAPL4 soil moisture datasets using the coupling stochastic model (Roundy and Wood, 2015) will illuminate the relationship between coupling strength and drought prediction accuracy. This forthcoming study aims to determine if a stronger coupling strength can truly enhance drought forecasts and will provide insights to representation of L-A coupling within the datasets. This has the potential to improve drought forecasting and thereby enhancing our understanding of drought dynamics using L-A interaction and strengthening the capacity for effective drought preparedness and response.

## Data Availability

All data mentioned in the dataset section of this paper are publicly available on their respective websites. In addition, the merged dataset for CTP and HI can be accessed via the following: Makhasana, P., J. Roundy, J. A. Santanello, P. M. Lawston-Parker (2024). Triple Collocation based Merged Dataset for Convective Triggering Potential (CTP) and Humidity Index (HI), HydroShare, https://doi.org/10.4211/hs.90bf9b575b684c849e617f620c2d63fb



**Author Contributions**

Payal R. Makhasana led the development of the Triple Collocation based merging dataset, conducted the analysis on coupling strength, and was primarily responsible for writing the original draft of the manuscript. Joseph A. Santanello and Patricia M.
Lawston-Parker contributed to the refinement of the study design, provided critical reviews, and edits to the manuscript, enhancing its intellectual content. Joshua K. Roundy developed the CTP-HI framework-based coupling classification code and participated in manuscript editing. Joseph A. Santanello, Patricia M. Lawston-Parker, and Joshua K. Roundy, played a significant role in shaping the research direction and methodology, ensuring the rigor and accuracy of the work presented.

**Competing interests**

The authors have no competing interests to declare.

**Acknowledgment**

We acknowledge the generous support and funding the NASA Soil Moisture Active-Passive Mission Science Team Program provided under grant NNH19ZDA001N-SMAP.

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
