# Peer review of "Deducing Land-Atmosphere Coupling Regimes from SMAP Soil Moisture"

_Hydrology and Earth System Sciences, 2024_

## Author Comment (AC1)

**Note: All revisions are highlighted in red, while the text that was originally part of the manuscript is highlighted in green. The review comments are presented in black, and our justifications and explanations are written in blue.**

Journal: HESS

Title: Deducing Land-Atmosphere Coupling Regimes from SMAP Soil Moisture

Authors: Makhasana, Santanello, Lawston-Parker, and Roundy

The paper examines the land-atmosphere coupling strength obtained from combining SMAP L3 or, separately, L4 soil moisture data with estimates of the convective triggering potential (CTP) and humidity index (HI) for the lower troposphere. The CTP and HI estimates are from a merged dataset created by the authors using Triple Collocation from three major atmospheric reanalysis datasets. The authors find that the CTP and HI estimates from the merged reanalysis dataset outperform CTP and HI estimates obtained from the individual reanalysis datasets (when compared to reference CTP and HI derived from radiosonde observations and, separately, AIRS satellite retrievals). The authors further find that SMAPL4 demonstrates stronger persistence of the wet and dry coupling regimes as compared to SMAPL3 and suggest that SMAPL4 may offer a robust approximation when assessing land-atmosphere interactions.

Overall, the manuscript has the potential to be an important contribution, but in its current form it falls short, as outlined in the comments below.

I recommend that the manuscript be returned to the authors for MAJOR revisions.

Thank you for your detailed review and constructive feedback. We understand your concerns regarding the clarity and structure. Your comments are indeed relatable, and we agree that each element could be reorganized for better coherence and readability.

Comments:

1) It is unclear how the objective of the study relates to the results. The objective of the study is described as follows:

Lines 85-87: "The goal of this comparative study is to uncover how soil moisture, as detected through direct satellite observations and assimilated data products, influences L-A coupling strength across the globe."

Lines 470-471: "The goal of this analysis is to explore the role of soil moisture from SMAP in quantifying the L-A coupling strength across the globe."

The key finding of the paper, however, appears to be related to the *difference* in the estimated coupling strength between SMAPL3 and SMAPL4 (see comment 2). This result does not quite match the formulation of the objective. The objective suggests that we will learn "how soil moisture [] influences L-A coupling strength across the globe." But the results only compare the different coupling strength estimates obtained for SMAPL3 and SMAPL4. The results do not examine the role of soil moisture as such in determining coupling strength, nor do they validate the coupling strength estimates. If the difference between SMAPL3 and SMAPL4 soil moisture could be interpreted as the error in the soil moisture estimates, then the results would examine the impact of the *error* in soil moisture on the estimates of coupling strength (rather than the impact of *soil* *moisture* on coupling strength as claimed). However, for the obvious reason of SMAPL3 and SMAPL4 being derived from the same sensor, the difference between SMAPL3 and SMAPL4 is not a good estimate of the error in the soil moisture data.

Thank you for the valuable comment. We agree that the relationship between the study's objective and the results needs to be clarified. We have revised the abstract to better align the objective and outcomes of the study. Below is the revised abstract that more clearly aligns the study's objectives with its outcomes, addressing the concerns raised.

Lines (11-22): "This study aims to identify the significance of soil moisture in identifying L-A coupling strength within the Convective Triggering Potential (CTP) and Humidity Index (HI) framework. To address this, a consistent and reliable dataset of atmospheric profiles is created by merging CTP and HI using Triple Collocation (TC) with three reanalysis datasets. The merged CTP and HI product demonstrates enhanced performance globally as compared to the individual datasets when validated with radiosonde and satellite observations. This merged product of CTP and HI is then used to compare the L-A coupling strength based on Soil Moisture Active Passive Level 3 (SMAPL3) and SMAP Level 4 (SMAPL4) over two decades (2003-2022) where L-A coupling strength is defined as the persistence probability within the dry and wet coupling regimes. Results indicate that the persistency-based coupling strength is related to the ability of soil moisture to predict future atmospheric humidity and dry vs. wet coupling state. The coupling strength in SMAPL4 is consistently stronger than in SMAPL3 and is likely due to its reliance on a land surface model and reduced susceptibility to random noise. The difference in coupling strength based on the same CTP-HI underscores the importance of soil moisture data in estimating coupling strength within the CTP-HI framework."

2) One of the key findings appears to be that "SMAPL4 demonstrates stronger persistence of the wet and dry coupling regimes as compared to SMAPL3. [...] This suggests that SMAPL4's approach may offer a robust approximation when assessing land-atmosphere interactions, .." (Lines 21-24; see also Lines 489-497). The implication here is that SMAPL4 is somehow better than SMAPL3 for the purpose, but the rationale for this remains unclear. Just because SMAPL4 results in stronger coupling estimates does not make these stronger estimates more correct. There is some discussion about this in Lines 498-499:

"..the representation of strong coupling might not always accurately mirror the complexities of real-world environmental conditions..".  However, the above key finding in the Abstract and elsewhere read as if there is no such caveat.  And even if, hypothetically, the caveat could be ignored, the authors do not explain *why* they think that stronger coupling is likely to be a better estimate.

Thank you for pointing out this shortcoming in the manuscript. It was never our attention to infer that the stronger coupling was somehow "more representative of reality". We have revised the relevant sections to provide a more comprehensive explanation and address this short coming.

Lines (567-577): "Fig. 7c shows that differences between SMAPL3 and SMAPL4 coupling strength is mainly due to the shape of the distribution of soil moisture and its projection in the CTP-HI space. A minor difference in soil moisture distribution is observed in Fig. 7c when comparing SMAPL4 and SMAPL4_L3, though this difference is more notable in the joint probability space of CTP-HI-SM (Fig. 7b) and CTP-HI Framework (Fig. 7a). Soil moisture estimates from SMAPL3 tend to skew towards drier values, likely due to the retrieval of the topsoil layer, which tends to dry quickly after rainfall events. This skew is influenced by fixed conditions in the retrieval process such as the prescribed freeze/thaw condition and water phase in the retrieval process, as well as land surface characteristics like vegetation cover and soil properties. In contrast, SMAPL4, which also uses observations from the same satellite, employs a dynamic approach by incorporating model-based soil hydraulic parameters, providing a more detailed depiction of soil moisture variations across different landscapes. Studies by Reichle et al. (2017), Reichle et al. (2019), Reichle et al. (2021) have shown a reduced bias and expand dynamic range of surface soil moisture in SMAPL4 as compared to in-situ and previous version of SMAPL4. These differences highlight the varying methodologies and characteristics of SMAPL3 and SMAPL4, resulting in distinct soil moisture estimates."

3) Related to comment 2): The authors suspect that the larger number of samples available from SMAPL4 explains the stronger coupling estimates (Lines 491-494).  Since the strength of the coupling is measured by the *persistence* of the coupling regime, I am not surprised that SMAPL4 results in stronger coupling estimates, simply because temporal auto-correlation of SMAPL4 soil moisture estimates is much higher than that of SMAPL3 estimates, owing to the fact that SMAPL4 soil moisture is partly derived from a land surface model and therefore less subject to random noise.  This explanation is perhaps hidden in the authors' language about "the unique characteristics of the SMAPL4's [sic] assimilation process."  (Lines 22-23)  However, the link is not obvious and should be examined and discussed more explicitly.

Thank you for your insightful comment. We agree that the link between stronger auto-correlation due to the land surface model in SMAPL4 likely plays a role in the stronger

coupling estimates and should be discussed more explicitly. Below, we have revised the discussion section to include a more detailed explanation of this relationship.

Lines (555-563):

"The higher coupling strength observed in SMAPL4 is attributed to the reliance on a land surface model and assimilation process. The constraints of a deterministic model structure based on fixed equations and inputs makes a land surface model less susceptible to random noise and thus creates a higher autocorrelation compared to SMAPL3. While this is a unique characteristic of SMAPL4's assimilation process, it is important to remember that the strong coupling may not accurately mirror the complexities of real-world conditions (Van Vuuren et al., 2012). The accuracy of the stronger coupling in SMAPL4 is difficult to quantity due to the scarcity of in-situ observations across the globe where simultaneous atmospheric profile and soil moisture measurements can be obtained. The limitation of comprehensive ground-based observations poses a significant challenge in validating the true representation of coupling and understanding the intricate interplay between soil moisture and atmospheric conditions (Santanello et al. 2018; Beamesderfer et al. 2022)."

4) Related to comment 3):  Figure 7 and Line 430: "The SMAPL4 dataset, with its higher number of wet regime classifications, demonstrates a greater likelihood of days being categorized as wet."  The logic here seems backward to me.  On average, SMAPL4 is wetter than SMAPL3, which is a consequence of the different approaches to soil moisture estimation in L4 and L3.  (It is unfortunate that the two estimates, despite coming from the same project, differ in their climatology, but such is the state of our knowledge of soil moisture.)  This climatological difference is clearly shown in Figure 7c.  But has this climatological difference been accounted for in the choice of CTP-HI classification parameters?  The manuscript is not clear about this.  The results (Fig 7a) suggest that the climatological difference between SMAPL4 and SMAPL3 is not considered in the CTP-HI classification.  If so, then it is not surprising that SMAPL4 leads to more "wet regime" classifications.  This needs to be examined further and clarified in the manuscript.

We agree that the description initially presented was not clear. The CTP-HI space is uniquely classified for each data set and the classification process fully accounts for the difference in climatology between the two data sets. We have revised the methodology section to make this explicit.

Lines (139-141): "The classification process is done for each dataset of CTP-HI-SM at the grid scale thus allowing for the coupling classification to account for the difference in climatology across datasets and regions around the globe, overcoming a limitation of the original Findell and Eltahir (2003) framework (Ferguson and Wood, 2011)."
The text around Fig. 7 was pointing out the fact that this difference in the soil moisture distribution (not its climatology) projected on to the CTP-HI space creates a greater persistence in wet coupling. We have revised the text describing Fig. 7 to better clarify this.

Lines (469-480): "Given that the classification algorithm accounts for climatological difference in the soil moisture datasets, the difference stems from the shape of the soil moisture distribution and its projection on to the CTP-HI space. The SMAPL3 dataset shows a tendency for observations to skew toward the lower end of the soil moisture spectrum, while the SMAPL4 tends to exhibit a clustering of observations in the mid-range, between 0.4 to 0.8. These differences in both the soil moisture distribution and its projection in the CTP-HI space affect the classification of coupling regimes and therefore the coupling strength of the timeseries. Fig. 7(d) depicts the daily coupling classification for an arbitrary month (May 2010). The SMAPL4 dataset, with its higher number of wet regime classifications, demonstrates a greater likelihood of days being categorized as a wet regime. This is evidenced in the time series, where most days are classified under wet conditions in SMAPL4, in contrast to the SMAPL3 dataset, which indicates more days in an atmospherically controlled regime. Sample size has a small impact on the classification, with only two days being different between SMAPL4_L3 and SMAPL4. This difference underscores the influence of soil moisture on the daily classification of L-A coupling within the CTP-HI framework, even though it is not directly used in creating daily timeseries."

5) In Line 12 and elsewhere, the authors state that they "examine the persistence of dry and wet coupling regimes over two decades (2003-2022)."  But SMAP data are available from April 2015 only.  Are the coupling strength results for the period starting April 2015?  This is very unclear in the manuscript.

Thank you for your insightful comment. The nuance of how soil moisture is used within the classification of the CTP-HI space is often a point of confusion. We have revised methodology section to better detail the role of soil moisture in the classification of the CTP-HI space and the development of the daily coupling classification. The methodology is now divided into three subsections: 2.1.1 Classification Input Variables, 2.1.2 Classification of the CTP-HI Space, and 2.1.3 L-A Coupling Strength. This includes adding a middle panel to Figure 1 that shows how soil moisture is utilized within the classification of the CTP-HI space.

Lines (116-123): Section 2.1.2 - "The classification process relies on daily values of the early morning estimates of CTP, HI and SM over a classification period. In this work, the classification period was selected as April 2015 to December 2022, to be consistent with the SMAP observational record. An example of the joint probability space, with the CTP in the x-axis, the HI in the y-axis, and the SM averaged over bins in the CTP-HI space is given in the middle panel of Fig. 1. This joint probability space is then used to define L-A coupling regimes within the 2-dimensional CTP-HI space based on the distribution of soil moisture. This is done by comparing the soil moisture in each bin to the climatological soil moisture using the two-sample Kolmogorov-Smirnov test. Bins with soil moisture distributions significantly wetter than the climatological distribution are classified as a wet regime bin, while those with significantly drier soil moisture distributions are classified as a dry regime bin."

Line (95):

[Figure]

**Figure 1:** A visual representation of the Soil Moisture (SM), Convective Triggering Potential (CTP), and Humidity Index (HI) on a thermodynamic diagram, along with a joint probability space of CTP-HI-SM at a specific grid location (38.89, -115.59). The CTP-HI framework is depicted after applying a two-sample Kolmogorov-Smirnov test on the joint probability space of CTP-HI-SM for the classification period from April 2015 to 2022.

Lines (143-150): Section 2.1.3 – "Once the CTP-HI space is classified based on estimates of morning observations of CTP, HI, and SM, a daily coupling timeseries can be generated. The daily coupling is determined by mapping the CTP and HI values for a day onto the classified CTP-HI space (right panel Fig. 1). For example, if the CTP and HI for a particular day map to a wet coupling regime, then that day is classified as a wet coupling regime day. This process is repeated for every day where there is an estimate of CTP and HI. Since the process for determining the daily coupling regime does not require the SM variable, the coupling timeseries can extend beyond the availability of SM data if there are CTP and HI data. Therefore, even though the CTP-HI space was classified on data from 2015-2022, the time series of daily coupling was extended to 2003 based on the availability of CTP and HI data from remote sensing."

6) The explanation of the methodology CTP-HI-SM classification approach should be improved. For example, Fig 1 talks about the "CTP-HI-SM space", but it remains unclear how SM enters the graphic on the right. In this graphic, CTP is on the abscissa and HI on the ordinate. But is SM shown in the shading? This is left to the reader's imagination. Is the CTP-HI-SM space in the right-hand graphic assembled by aggregating over space and/or time? Related to this, the text in Lines 111-115 is a bit too brief to be understood without referring to Findell and Eltahir (2003) and/or Roundy et al 2013.

Thank you for your feedback. As mentioned above we completely revised the section on the CTP-HI-SM classification. This includes dividing it into three subsections: 2.1.1 Classification Input Variables, 2.1.2 Classification of the CTP-HI Space, and 2.1.3 L-A Coupling Strength and adding a middle panel to Figure 1 that shows how soil moisture is utilized within the classification of the CTP-HI space.

7) The selected references are often inappropriate.

Thank you for your observations regarding the selected references. We have verified all the references again and made the following adjustments:

- The Triple Collocation references (Lines 76-77) consist of three recent applications, at least two of which are highly specific regional studies. Gruber et al (2017), which is cited elsewhere, would be more appropriate, or perhaps better still would be the review paper by Gruber et al (2020) doi:10.1016/j.rse.2020.111806 and/or the seminal paper by Stoffelen et al (1998) doi:10.1029/97JC03180.

Lines (69-71): "However, the Triple Collocation (TC) method has emerged as an invaluable technique for estimating error variances within datasets, as evidenced by research from Gruber et al., (2017), Gruber et al., (2020), Stoffelen, (1998), and Saha et al. (2020)."

- Line 223: Ochege et al (2017) is not appropriate as the introductory reference for MERRA-2. The relevant reference is Gelaro et al (2017), which appears in the following line.

Lines (252-253): "MERRA2 provides 6-hourly observations with an approximate spatial resolution of 0.5°x0.625° and includes 72 hybrid pressure levels ranging from the surface to 0.01hPa Gelaro et al. (2017)."

- Line 229: Centella-Artola et al (2020) is not appropriate as the introductory reference for CFSR. The relevant reference is Saha et al (2010, which appears a few lines later.

Lines (259-260): "It covers the period from 1979 to the present. It provides six-hourly variables estimations, including 64 atmospheric levels at a 0.5° x 0.5° horizontal resolution (Saha et al. 2010)."

Besides, following references were not placed properly as well,

Lines (267-268): ERA5 provides hourly land and atmospheric climate variable estimations at approximately a 31 km spatial resolution and 137 levels from the surface to 80 km (Bell et al., 2021).

Lines (580-582): "For instance, Xu (2020) has concluded that SMAPL4 surface soil moisture product is more accurate, with lower errors (ubRMSE < 0.04 m³/m³), compared to the SMAPL3 product (~0.06 m³/m³)."

Lines (585-587): "Reichle et al., (2017) have shown that version 4 SMAPL4's bias is significantly reduced as compared to version 3, exhibiting a 46% decrease in surface soil moisture uncertainty."

8) The use of land surface observations (soil moisture, snow, precipitation) is quite different across the three reanalysis datasets used here. This information should be included in the brief introductions of the reanalysis datasets (sections 3.1.1, 3.1.2, and 3.1.3). In addition to the screen-level obs, ERA5 also assimilates soil moisture retrievals from spaceborne scatterometers, which is not mentioned in section 3.1.3. CFSR and MERRA2, on the other

hand, use observation-based precipitation to force the land model within the reanalysis system.

Thank you for the comment. As suggested, we have revised sections 3.1.1, 3.1.2, and 3.1.3, to incorporate the necessary details:

Lines (248-279):

"3.1.1 The Modern-Era Retrospective Analysis for Research and Application, version 2 (MERRA2)

NASA's Global Modelling and Assimilation Office (GMAO) developed MERRA2 as an atmospheric reanalysis dataset, employing the Goddard Earth Observing System (GEOS) Atmospheric General Circulation Model (AGCM). The AGCM is a sophisticated numerical model that simulates the Earth's atmospheric processes, providing a comprehensive framework for understanding climate dynamics and variability. MERRA2 provides 6-hourly observations with an approximate spatial resolution of 0.5°x0.625° and includes 72 hybrid pressure levels ranging from the surface to 0.01hPa (Gelaro et al. 2017). The data assimilation system of MERRA2 utilizes the 3D-var algorithm and spans from 1980 to the present. Gelaro et al. (2017) describe how the dataset incorporates observation-based precipitation to force the land model, ensuring realistic precipitation inputs, along with advancements and improvements made in the system.

3.1.2 The Climate Forecast System Reanalysis (CFSR)

The Climate Forecast System Reanalysis (CFSR) is developed by the National Center for Environmental Prediction (NCEP). It covers the period from 1979 to the present. It provides six-hourly variables estimations, including 64 atmospheric levels at a 0.5° x 0.5° horizontal resolution (Saha et al. 2010). Operating as a global coupled atmosphere-ocean-land surface-sea ice system, CFSR incorporates satellite radiance data and employs the Integrated Forecasting System (IFS) Cycle 41r2 with the 3D-var data assimilation system. Observations are carefully considered for each component during the assimilation process of the CFSR dataset, however CFSR uses observation-based precipitation to force the land model, enhancing precipitation accuracy, as highlighted in Saha et al. (2010).

3.1.3 European Centre for Medium-Range Weather Forecast (ECMWF) Reanalysis v5 (ERA5)

ERA5, the fifth ECMWF reanalysis data of global climate, is accessible from January 1959 to the present and produced by the Copernicus Climate Change Service (C3S). ERA5 provides hourly land and atmospheric climate variable estimations at approximately a 31 km spatial resolution and 137 levels from the surface to 80 km (Bell et al., 2021). It employs the Integrated Forecasting System (IFS) Cycle 41r2 and assimilates satellite and in-situ observations. ERA5 includes advanced screen-level assimilation for 2m temperature and relative humidity components, where the soil moisture is nudged to better match the 2-

meter observations. ERA5 assimilates soil moisture from spaceborne scatterometers and integrates various precipitation data sources, improving soil moisture and precipitation estimates. Hersbach et al. (2020) compared ERA5 with radiosonde data and showed temperature, wind, and humidity improvements in the troposphere for the latest version.

Differences in land surface observations among these datasets can impact atmospheric variables and introduce biases. Soil moisture influences evaporation and humidity, while observation-based precipitation enhances land model accuracy, influencing atmospheric moisture and stability. ERA5 benefits from direct soil moisture assimilation, which potentially reduces bias. In contrast, MERRA2 and CFSR use observation-based precipitation to force their land models and rely on model-generated soil moisture. This approach can introduce bias in temperature and humidity profiles due to uncertainties in the model soil moisture. Understanding these differences is essential for interpreting reanalysis data."

9) Section 3.2 is missing information on how the SMAPL3 quality flags are used, if at all. Are SMAPL3 screened when they are flagged for less-than-optimal quality?

Thanks for pointing out this shortcoming. We have added text in the methodology about our choice of quality flags as well as added some text in the discussion section on how this choice may have impacted the results.

Lines (326-327): "All available SMAPL3 data are used without filtering based on the quality flags in order to maintain a larger dataset for a comprehensive analysis."

We also add some context as to how this choice of not filtering the SMAPL3 data based on quality flags could impact the results of this study.

Lines (548-550): "One possible reason for the weaker coupling strength in SMAPL3 is that all available SMAPL3 data points were used to maintain a larger dataset without considering the quality flags."

10) Figures 4, 5, and 9: Since the individual regions cover very different total areas, it is difficult to derive the skill or coupling metrics for the entire globe from visual inspection. In these bar charts, I suggest adding a 7-th group of bars with the metrics for the entire globe.

Thanks for this suggestion. We have implemented a global bar to help provide context to the regional metrics.

Lines (391-393):

[Figure]

"Figure 4: Bar plot of performance metric and intercomparison of merged data and reanalysis with radiosonde observation from IGRA2 in different region and globally for (a) CTP and (b) HI."

Lines (394-396):

[Figure]

"Figure 5: Bar plot of performance metric and intercomparison of merged data and reanalysis with satellite remote sensing from AIRSv7 in different region and globally for (a) CTP and (b) HI."

Lines (514-516):

[Figure]

"Figure 9: Comparative analysis of Land-Atmosphere average coupling strength using different soil moisture from SMAPL3, SMAPL4_L3, and SMAPL4 data across various region and globally."

11) There are many typos and grammatical errors throughout the manuscript.  See "Editorial comments" below for a sampling.  While the impact of these errors on the readability of the paper is relatively small, they reflect poorly on the quality of the study.  Have the senior coauthors (who are all native speakers) proofread the paper?

We apologize for any inconvenience caused by the typos and grammatical errors in the manuscript. We have thoroughly reviewed the paper multiple times and made the necessary corrections to enhance the readability and overall quality of the paper. We believe these revisions will significantly improve the clarity and presentation of our study.

Minor comments:

Thank you for your detailed review and the minor comments provided. We have addressed each comment:

1. a) Line 17: "Despite significant correlated errors within the individual reanalysis datasets, .." Do you mean "Despite significant error correlations across the individual reanalysis datasets, .."?  Or are you referring to "temporally (auto-)correlated errors within the individual reanalysis datasets"?  Please clarify the exact error correlation implied here.

   To clarify, we were referring to significant error correlations across the individual reanalysis datasets. Specifically, Figure 3(a) illustrates the correlated errors between IGRA2 and reanalysis datasets.
   However, we have revised the abstract, and the statement referring to "significant correlated errors within the individual reanalysis datasets" has been removed in the revised version.

2. b) "HI measures low atmospheric moisture levels" (Line 52) This is not about "low [as opposed to high] moisture levels" (that is, dry vs. wet air), right? Do you mean "HI measures moisture levels in the lower troposphere"? Or, perhaps better: "HI measures moisture *content* in the lower troposphere". ("Levels" is easily confused with the "model levels" used in atmospheric models.)

To clarify, the intent was to describe the measurement of moisture content in the lower troposphere. We have removed the term "levels" to avoid confusion.

Lines (49-50): "HI quantifies moisture content in the lower troposphere."

3. c) Line 134 says that "the atmospheric controlled and transitional regimes are combined into one regime", but Fig 7 then distinguishes between the transitional and "atmospheric controlled" regimes. This is contradictory.

Thank you for your observation.
Line (481)

[Figure]

Lines (460-462): "Fig. 7(a) illustrates the classified CTP-HI space based on SMAPL3, SMAPL4_L3, and SMAPL4 datasets. The coupling regimes are clearly distinguished within the CTP-HI framework across the datasets, highlighting the variations and interactions between soil moisture and atmospheric conditions."
Lines (137-139): "To simplify the analysis and to emphasize the crucial role that soil moisture plays in defining the dry and wet regimes, the atmospherically controlled and transitional regimes are merged into a single category termed atmospherically controlled for this analysis."

4. d) Line 115, equation (1): Missing plus sign after "a_i"?

Line (177):

$$\theta_i = a_i + b_i\theta + \varepsilon_i$$

5. e) Lines 168-169, equations (2) and (3): Missing definition of symbols \mu and \sigma

   Line (192):
   "In the above step, μ represents the mean and σ represents the standard deviation of the respective datasets."

6. f) Line 176: "..differences between two variables.." Do you mean "..differences between two datasets..?" (That is, differences in the estimate of, say, CTP from different reanalysis datasets.)

   You are correct.
   (Line 199)"...differences between two datasets over the study area."

7. g) Lines 189-190: "Additionally, the study comprises a 30-day centered window (15 days on either side of the compound event) that removes the effect of seasonality." This information comes too late and should be moved up. It is fundamental to the success of Triple Collocation that the seasonal cycle is removed from the data first.

   Thank you for pointing out the importance of removing the seasonal cycle prior to the triple collocation analysis. Taking a 30-day centered window is indeed the first step in our methodology. We included this information later in the text while summarizing our approach to handling seasonality. Now we have moved this information earlier in the document to emphasize its fundamental role in the analysis. (Lines : 170-172)

8. h) Line 221: MERRA-2 is an atmospheric reanalysis, and the variables of interest for this paper are estimates of atmospheric conditions (CTP and HI). Therefore, description of MERRA-2 should mention the GEOS AGCM, not just the Catchment land model. In the context of the present study, the Catchment model is much more relevant as the land surface model underpinning the SMAPL4 land data assimilation system.

   This version acknowledges the GEOS AGCM:

   Lines (249-252):
   "NASA's Global Modelling and Assimilation Office (GMAO) developed MERRA2 as an atmospheric reanalysis dataset, employing the Goddard Earth Observing System (GEOS) Atmospheric General Circulation Model (AGCM). The AGCM is a sophisticated numerical model that simulates the Earth's atmospheric processes, providing a comprehensive framework for understanding climate dynamics and variability."

9. i) Line 276: The "resolution" of the (enhanced) SMAPL3 data is not 9 km. Unfortunately, there is some misleading information on the NSIDC web

documentation. The true resolution of the "enhanced" L3 retrievals is closer to ~30 km.

Lines (312-317):

"Enhanced SMAP Level 3 (SMAPL3) products, derived from the foundational Level 1 and 2 data, provide standardized, gridded global soil moisture (O'Neill et al. 2021) at 9 km resolution with the capability to observe the global Earth surface every 2-3 days. While the Enhanced SMAP Level 3 is provided at a 9 km resolution, it should be noted that the native radiometer footprint is at ~36 km and the brightness temperatures are interpolated to the 9 km resolution using an optimally localized average method."

10. j) Figures 3, 4, 5, 7, and 8 are missing units for CTP and HI.

Change have been applied in respective figures (Fig 1, Fig. 3, Fig. 4, Fig. 5, Fig. 6, Fig. 7 , Fig. 8, and Fig. 9).

11. k) Figure 3a is a scatterplot of *differences* (or errors vs obs). Accordingly, x- and y-labels should read "CFSR minus IGRA", "MERRA-2 minus IGRA", and "ERA5 minus IGRA".

In Figure 3a, we have revised the figure to clearly indicate that it represents the correlated errors between the datasets, not the reanalysis datasets themselves. This clarification should help avoid confusion regarding the labels and the data being presented.

Lines (335-338)

"In Fig. 3(a), the scatterplots show the correlated errors between different reanalysis datasets (MERRA2, CFSR, and ERA5) with respect to the IGRA2 observations at a location in Kansas, United States (coordinates 39.96, -95.26). For instance, MERRA2 vs CFSR represents the errors in MERRA2 plotted against the errors in CFSR, both with respect to the IGRA2 observations."

12. l) The numbers in Table 2 should all have the same number of decimals. I think two decimals (or integer values in percentage terms) would be sufficient and much easier to read. (A disclaimer could be added that the percentage values may not add to 100 because of roundoff error.)

Lines (369) :

| Weight distribution across the region  for CTP (J/Kg) | | | | | | |
|---|---|---|---|---|---|---|
| Region | NAM | SAM | AFR | EUR | ASA | AUS |
| MERRA2 | 0.27 | 0.27 | 0.32 | 0.35 | 0.28 | 0.30 |
| CFSR | 0.30 | 0.30 | 0.32 | 0.34 | 0.30 | 0.30 |
| ERA5 | 0.43 | 0.43 | 0.35 | 0.31 | 0.42 | 0.40 |
| Weight distribution across the region  for HI (°C) | | | | | | |
| Region | NAM | SAM | AFR | EUR | ASA | AUS |
| MERRA2 | 0.33 | 0.29 | 0.35 | 0.37 | 0.35 | 0.31 |
| CFSR | 0.31 | 0.31 | 0.33 | 0.34 | 0.30 | 0.34 |
| ERA5 | 0.36 | 0.40 | 0.32 | 0.29 | 0.35 | 0.36 |

13. m) Line 339: "The data is [sic] merged following equation (10)." This equation provides the objective function for determining the optimal weights. It is not the equation used to merge the datasets, which is presumably:   CTP_merged = w_M2 * CTP_M2 + w_ERA5 * CTP_ERA5 + w_CFSR * CTP_CFSR.

Lines (209-211):
The merged CTP and HI is then calculated using the weighted sum of the individual datasets:

$$CTP_{merged} = w_{MERRA2}.CTP_{MERRA2} + w_{CFSR}.CTP_{CFSR} + w_{ERA5}.CTP_{ERA5} \quad (10a)$$
$$HI_{merged} = w_{MERRA2}.HI_{MERRA2} + w_{CFSR}.HI_{CFSR} + w_{ERA5}.HI_{ERA5} \quad (10b)$$

14. n) Line 350: "discrepancies" with respect to what?

"…discrepancies with observational data."

15. o) Line 375: Clarify if ~1:30AM is the *local* overpass time.

Noted and change have been applied accordingly

16. p) Line 378: "Fig 4" seems to be the wrong reference here

Thanks for pointing out, it supposed to be Table 2.

17. q) Line 402: "a more reliable predictor" - more reliable than what?

Lines (447-448):
"Fig. 6(b) and 6(c) show the average lag correlation out to 10 days over the contiguous US and indicate that soil moisture has a stronger predictive influence on CTP and HI as shown by the larger magnitudes of correlations over positive lag."

18. r) Figure 7c,d: What do you mean by "Saturate Soil Moisture"? Do you mean "soil moisture in units of relative saturation" (or "wetness" units)?

Thank you for your question. In Fig. 7c,d, "Soil Moisture" has been redefined as "Relative Saturation Soil Moisture" for clarity. This term refers to soil moisture

expressed in units of relative saturation, which is presented as a dimensionless ratio.

19. s) Figure 7b: What exactly do you mean by "Joint Probability of CTP-HI-SM space"? How is the graphic showing a "probability"?

Thank you for your question. In Fig. 7b, the term "Joint Probability of CTP-HI-SM space" is used to describe the relationship between soil moisture (SM), Convective Triggering Potential (CTP), and Humidity Index (HI). We analyze historical observations to identify patterns in how soil moisture influences the lower atmosphere as represented by CTP and HI.

Although we refer to it as a joint probability space, we are not looking at probabilities in the strict statistical sense. Instead, we are examining the responses of soil moisture at specific locations for morning observations of CTP and HI. The graphic illustrates the frequency of occurrences of various combinations of CTP, HI, and SM, providing insight into the interactions between these variables. We have added some text to better describe this.

Lines (463-465):
"In Fig. 7(b), the joint probability of CTP-HI-SM space illustrates the bin average SM within the CTP-HI space based on historical observations and helps to identify patterns in the CTP-HI-SM relationship."

20. t) Lines 443-446: It would be helpful to insert "Fig 8a" and "Fig 8b" here to help the reader identify the specific part of the graphic that illustrates the statements made here.

Lines (492-495):
As indicated in Fig. 8(a), all soil moisture datasets show a weak relationship between average coupling strength and average lag correlation for CTP as indicated by a lower $R^2$ (explained variance by the regression line) and the shallow slope of the regression line. In contrast, Fig. 8(b) shows the SM-HI relationship is stronger, with higher R2 and more pronounced nonlinear relationship.

21. u) Figure 8: It would be helpful to add "CTP" in the top row and "HI" in the bottom row of the graphic.

Change have been applied (Line 501):

[Figure]

22. v) Line 474: Should "Figs. 7 and 8" read "Figs. 6 and 8"??

   Agreed. Change have been applied.

23. w) Lines 519-520: "For instance, Xu (2020) has shown that SMAPL4's bias is significantly reduced,…" Reduced with respect to what?

   To clarify, "… with respect to SMAPL3…"

   Lines (580-582): For instance, Xu (2020) has concluded that SMAPL4 surface soil moisture product is more accurate, with lower errors (ubRMSE < 0.04 m³/m³), compared to the SMAPL3 product (~0.06 m³/m³).

24. x) Lines 521-522: ".. which showed that SMAPL4 captures spatial and temporal soil moisture variations more reliably across the United States." More reliably than what?

   To clarify, "…than Advanced Microwave Scanning Radiometer (AMSR2) soil moisture…."

   Lines (583-585): This enhanced accuracy has been corroborated by Zhang et al. (2017), which showed that SMAPL4 captures spatial and temporal soil moisture variations more reliably as compared with Advanced Microwave Scanning Radiometer (AMSR2) across the United States.

25. y) Lines 527-528: "The SMAP provides enhanced depiction of L-A coupling through dynamic soil moisture data, offering improved drought monitoring and weather prediction." This statement is not supported by the results or a reference.

   We have removed this statement while revising

26. z) Lines 543-544: "Despite this, the merged dataset still demonstrates a more accurate reflection of in-situ and satellite observations of CTP and HI,.." More accurate than what??

To clarify, Lines (530-532):

"Despite this, the merged dataset demonstrates a more accurate reflection of in-situ and satellite observations of CTP and HI compared to individual datasets, thus providing a temporal and spatially consistent dataset for analysing L-A coupling."

Editorial comments:

Thank you for your suggestion.

Lines 47-49: Delete "to illuminate the L-A coupling" Change has been applied.

Line 127: Capitalize "Hi" - - > "HI" Change has been applied.

Line 176: Equations (4)-(6) use (curly) "braces" not "brackets". Change has been applied.

Equations (4)-(6) with the curly braces:

$$\varepsilon^2_{MERRA2} = \{(\theta_{MERRA2} - \theta'_{CFSR})(\theta_{MERRA2} - \theta'_{ERA5})\} \qquad (4)$$
$$\varepsilon^2_{CFSR} = \{(\theta_{CFSR} - \theta'_{MERRA2})(\theta_{CFSR} - \theta'_{ERA5})\} \qquad (5)$$
$$\varepsilon^2_{ERA5} = \{(\theta_{ERA5} - \theta'_{CFSR})(\theta_{ERA5} - \theta'_{MERRA2})\} \qquad (6)$$

The following sentences are a sampling of the grammatical errors or otherwise difficult-to-read sentences mentioned above:
Thank you for pointing out the grammatical errors and difficult-to-read sentences. Below are the revised sentences for better clarity and readability:

Line 20: "a higher lag-correlation between soil moisture and the CTP-HI metrics contribute to the persist coupling behaviour"
We have removed above statement while revising abstract.

Lines 78-79: "Therefore, using the TC method to merge reanalysis data sets of CTP and HI based has the potential to provide."
Lines (73-74): "Consequently, the TC method is an ideal choice to create a more robust merged CTP and HI metric for analysis of L-A coupling strength."

Lines 108-111: "In the revised CTP-HI framework [..], the interplay between soil moisture and atmospheric conditions is distinguished into four specific coupling regimes: wet coupling, dry coupling, transitional, and atmospherically controlled; and

summarize the complex relationship between soil moisture content and the feedback from the land to the atmosphere in a generalized context."

Lines (137-139): "To simplify the analysis and to emphasize the crucial role that soil moisture plays in defining the dry and wet regimes, the atmospherically controlled and transitional regimes are merged into a single category termed atmospherically controlled for this analysis."

Lines 200-202: "To assess the performance of merged CTP-HI the analysis also includes Atmospheric Infrared Sounder Version 7(AIRSv7) satellite remote sensing and radiosonde observations from Integrated Global Radiosonde Archive Version 2 (IGRA2)."

Lines (221-223): "Satellite remote sensing and in-situ data are used to assess the performance of the merged CTP-HI dataset. Specifically, CTP and HI are calculated using data from the Atmospheric Infrared Sounder Version 7(AIRSv7) as well as radiosonde observations from Integrated Global Radiosonde Archive Version 2 (IGRA2)."

Lines 459-460: "However, when considering the impact of sample size, the difference in coupling strength is dimensioned."   [What does this mean??]

Lines (508-509): "The variation in coupling strength becomes noticeable when the sample size is considered."

Lines 506-507: "In synthesizing the comparison between SMAPL3 and SMAPL4, as depicted in Fig. 7c, highlights the differences in soil moisture representation arise mainly from their distinct constraints and processing methodologies."

Lines (567-568): "Fig. 7c shows that differences between SMAPL3 and SMAPL4 coupling strength is mainly due to the shape of the distribution of soil moisture and its projection in the CTP-HI space."

Line 555: "Such stronger persistence of wet and dry coupling regimes, as observed in SMAPL4 is not only a result of a greater number of observations, but it possibility due to the distinctive assimilation techniques employed in the SMAPL4 dataset."

Lines (604-605): "The increased coupling strength in SMAPL4 may result from SMAPL4's reliance on a land surface model which reduces susceptibility to random noise compared to SMAPL3."

---

## Author Comment (AC2)

**Note: All revisions are highlighted in red, while the text that was originally part of the manuscript is highlighted in green. The review comments are presented in black, and our justifications and explanations are written in blue.**

This study combines different soil moisture and atmospheric data products to evaluate land-atmosphere coupling within the classical CTP-HI framework. The scientific approach and methods used are sound, and the results will be of interest to the land-atmosphere interactions community. Although the scientific elements are strong, it was at times difficult to clearly grasp what they authors were trying to achieve and communicate. I believe that the paper could benefit from some minor revisions that would help strengthen the narrative and better highlight its key points. These are summarized by my comments on the abstract but extend to the rest of the paper's discussion as well:

Thank you for your detailed review and constructive feedback on our manuscript. We understand your concerns regarding the clarity and structure. Your comments have helped us revise and reorganized the manuscript for better coherence and readability.

**Abstract Revisions**

The abstract is rather long and fails to adequately set up the study's main goals. Many elements are introduced rather haphazardly and readers may struggle to connect the dots. Some sentences are also repetitive, leading to greater confusion. For example, the abstract begins by stating that "this research assesses the impact of different soil moisture datasets on the classification and distribution of L-A coupling regimes." Then, the following sentence states that the goal is to "examine the persistence of dry and wet coupling regimes... exploring how soil moisture influences coupling classification." Although these sentences are certainly related, it's not clear then whether the main goal is to assess differences between the SMAP data products or to more broadly evaluate soil moisture coupling. The term "persistence of dry and wet coupling regimes" is also introduced without much context, though it seems like "persistence" is a central concept to the study and how the authors are thinking about coupling. Despite the title and first few sentences of the abstract setting up soil moisture and the different SMAP data products as the main focus of the study, the bulk of the abstract is spent discussing the need for consistent and unbiased observations of the atmospheric state and the merged reanalysis product the authors created.

After rereading the abstract a few times and digesting the rest of the manuscript, here are my suggestions for restructuring:

Thank you for your detailed comments about the abstract. We agree that the original abstract was disconnected and failed to convey the work in the paper. We apologize for the inconvenience. We have taken your suggestions and completely revised the abstract to better reflect the goals, work and conclusions of the paper. The detailed revised abstract is presented below.

1.  Start off with: "In recent years, there has been a growing recognition of the significance of Land-Atmosphere (L-A) interactions and feedback mechanisms and their importance for weather and climate prediction." (no change)

    Revision (no change):

2.  Lead into a sentence explaining why L-A coupling regimes are useful/important/of interest: e.g., "L-A coupling regimes are a useful framework for understanding…"

    Revision - Lines (9-11)

3.  Then set up the study's focus and contribution with respect to data products: e.g., "Characterizing and studying L-A coupling regimes requires consistent and unbiased observations of surface conditions and the atmosphere…"

    Revision - Lines (12-18)

4.  Now state the main goal of the study, succintly in one sentence: e.g., "We compare the classification and distribution of L-A coupling regimes across different soil moisture datasets by computing the lag correlation between the SMAP Level 3 and Level 4 soil moisture products and Convective Triggering Potential (CTP) and Humidity Index (HI) from a merged reanlaysis product we develop."

    Revision - Lines (11-12)
    We are stating the main goal of the study before explaining the study's focus and contribution with respect to the dataset. This approach ensures that the reader understands the primary objective before delving into how the study contributes to the understanding and analysis of L-A coupling regimes through consistent and unbiased observations.

5.  1-3 sentences for findings: e.g., "We find that the persitence of dry and wet coupling regimes during the time period of the study can be understood through…"

    Revision - Lines (19-22)

6.  End with sentence on significance: "These findings lay the groundwork for understanding the sensitivity of drought evolution to soil moisture variations by gaining insight into the quantification of coupling strength, thereby providing critical insights for future drought modelling and prediction efforts." (second part of the sentence is repetitive and can be removed)

    Revision - Lines (22-24)

"**Abstract.** In recent years, there has been a growing recognition of the significance of Land-Atmosphere (L-A) interactions and feedback mechanisms in understanding and predicting Earth's water and energy cycles. Soil moisture plays a critical role in mediating the strength of L-A interactions and is important for understanding the complex and governing processes across this interface. This study aims to identify the significance of

soil moisture in identifying L-A coupling strength within the Convective Triggering Potential (CTP) and Humidity Index (HI) framework. To address this, a consistent and reliable dataset of atmospheric profiles is created by merging CTP and HI using Triple Collocation (TC) with three reanalysis datasets. The merged CTP and HI product demonstrates enhanced performance globally as compared to the individual datasets when validated with radiosonde and satellite observations. This merged product of CTP and HI is then used to compare the L-A coupling strength based on Soil Moisture Active Passive Level 3 (SMAPL3) and SMAP Level 4 (SMAPL4) over two decades (2003-2022) where L-A coupling strength is defined as the persistence probability within the dry and wet coupling regimes. Results indicate that the persistency-based coupling strength is related to the ability of soil moisture to predict future atmospheric humidity and dry vs. wet coupling state. The coupling strength in SMAPL4 is consistently stronger than in SMAPL3 and is likely due to its reliance on a land surface model and reduced susceptibility to random noise. The difference in coupling strength based on the same CTP-HI underscores the importance of soil moisture data in estimating coupling strength within the CTP-HI framework. These findings lay the groundwork for understanding the role of L-A interactions and drought evolution due to soil moisture variations, by providing insight into the quantification of coupling strength and its role in drought monitoring and forecast efforts."

**Other Clarifications**

1. **Timescales of Coupling (L393-394):** My understanding is that the CTP-HI framework is typically used to evaluate land-atmosphere coupling on diurnal timescales. In particular, the original Findell and Eltahir papers focus on how soil conditions influence the evolution of the early morning atmosphere. In this study, the authors evaluate coupling between soil moisture and the CTP-HI metrics by computing the lagged correlation over a 10-day average. In lines 393-394 the authors state this is because "reliable weather predictability is generally limited to 10 days." While this is true for numerical weather prediction, this argument seems less relevant for a LA study. I think this is an interesting aspect of the paper that could be expanded on. Please provide more context and justifiation for the timescale of coupling and maybe highlight some other works that have evaluated coupling over this timescale. I believe some of the cited works are relevant here; a review paper some of the authors were involved in (Santanello et al., 2018) also has an excellent discussion of this.

Thank you for your insightful comment  regarding the timescales of coupling. We recognize the need to provide a clearer justification for using a 10-day average in lag-correlation study. The primary reason behind our use of a 10-day average is to capture the persistence of soil moisture influences beyond a single diurnal cycle. Soil moisture anomalies can have prolonged impacts on atmospheric processes, and this extended timescale helps us understand the cumulative effects of soil moisture on land-atmosphere interactions. We have revised the manuscript to better articulate this rationale, highlighting the relevance of a 10-day period in capturing the persistence of soil moisture influence.

Lines (430-442): "To delve deeper into the noted differences in coupling strength, the lagged correlation between the three sets of soil moisture and the CTP and HI over 2015-2022 are analyzed. Lag correlation is employed to identify the relationship between soil moisture and future CTP and HI and vice versa. While previous work has discussed the potential for soil moisture predictability out to 60 days (Dirmeyer et al., 2018), this analysis uses a 10-day lag to capture the role of soil moisture in predicting the atmospheric state (CTP and HI) and the atmospheric state in predicting soil moisture on time scales relative to typical weather predictability. Within this setup, the ability of soil moisture to predict future CTP and HI is given as a positive lag correlation and the ability of the CTP and HI to predict future soil moisture is given as a negative lag. For both the CTP and HI, the correlation with soil moisture is negative due to the relationship between SM-CTP and SM-HI. Wet soil typically results in surface cooling when solar radiation is limited, leading to a more stable temperature profile in the lower atmosphere. This stability restricts vertical movement and consequently leads to a lower CTP, thus creating a negative correlation. HI, on the other hand, measures atmospheric moisture content. Higher HI values signify drier air, while lower values indicate moisture-rich air closer to saturation. High soil moisture enhances evaporation, which adds water vapor to the atmosphere, reducing the gap between temperature and dew point and thus lowering the HI, resulting in a negative correlation."

Lines (447-458): "Fig. 6(b) and 6(c) show the average lag correlation out to 10 days over the contiguous US and indicate that soil moisture has a stronger predictive influence on CTP and HI as shown by the larger magnitudes of correlations over positive lag. This is particularly noticeable for shorter lags, suggesting a more immediate impact of soil moisture on atmospheric stability and humidity. Conversely, the decrease in correlation magnitude with longer lags highlights the diminishing influence of L-A interaction over time. For the different datasets, SMAPL4 consistently shows higher correlations at all lag intervals for both CTP and HI. However, the sample size does play a role in this assessment as noticed by a decrease in the magnitude of correlation for the SMAPL4_L3. Despite this, the SMAPL4_L3 dataset still shows a higher magnitude in lag correlation as compared SMAPL3, particularly for CTP. This suggests that the assimilation of SMAP observations into a model, as in SMAPL4, may yield a stronger relationship in the temporal dynamics between the land surface and the atmosphere. In contrast, the pattern of a stronger L-A connection for SMAPL4 is less evident for HI."

Also, the outcome of the lag correlation analysis is already well mentioned in the conclusion section (lines 600-604)

2. **Time Period of Study (L290-292):** The abstract states that the study "examine the persistence of dry and wet coupling regimes over two decades (2003–2022), exploring how soil moisture influences coupling classification." This seems a little misleading given that SMAP is only available for the last eight years of that time period. I was a little confused by what "Since soil moisture measurements are only needed for the classification period, the

time series of coupling classiciation can still cover the entire time series of CTP-HI from 2003 to 2022." in Lines 290-292 meant. Should this instead say "Since soil moisture measurements are only needed for the lagged correlations?" Please explain and also be more careful about stating the time period of the results in the rest of the study.

Thank you for your insightful comment. The nuance of how soil moisture is used within the classification of the CTP-HI space is often a point of confusion. We have revised methodology section to better detail the role of soil moisture in the classification of the CTP-HI space and the development of the daily coupling classification. The methodology is now divided into three subsections: 2.1.1 Classification Input Variables, 2.1.2 Classification of the CTP-HI Space, and 2.1.3 L-A Coupling Strength. This includes adding a middle panel to Figure 1 that shows how soil moisture is utilized within the classification of the CTP-HI space.

Lines (116-123): Section 2.1.2 - "The classification process relies on daily values of the early morning estimates of CTP, HI and SM over a classification period. In this work, the classification period was selected as April 2015 to December 2022, to be consistent with the SMAP observational record. An example of the joint probability space, with the CTP in the x-axis, the HI in the y-axis, and the SM averaged over bins in the CTP-HI space is given in the middle panel of Fig. 1. This joint probability space is then used to define L-A coupling regimes within the 2-dimensional CTP-HI space based on the distribution of soil moisture. This is done by comparing the soil moisture in each bin to the climatological soil moisture using the two-sample Kolmogorov-Smirnov test. Bins with soil moisture distributions significantly wetter than the climatological distribution are classified as a wet regime bin, while those with significantly drier soil moisture distributions are classified as a dry regime bin."

Line (95):

[Figure]

Figure 1: A visual representation of the Soil Moisture (SM), Convective Triggering Potential (CTP), and Humidity Index (HI) on a thermodynamic diagram, along with a joint probability space of CTP-HI-SM at a specific grid location (38.89, -115.59). The CTP-HI framework is depicted after applying a two-sample Kolmogorov-Smirnov test on the joint probability space of CTP-HI-SM for the classification period from April 2015 to 2022.

Lines (143-150): Section 2.1.3 – "Once the CTP-HI space is classified based on estimates of morning observations of CTP, HI, and SM, a daily coupling timeseries can be generated. The daily coupling is determined by mapping the CTP and HI values for a day onto the classified CTP-HI space (right panel Fig. 1). For example, if the CTP and HI

for a particular day map to a wet coupling regime, then that day is classified as a wet coupling regime day. This process is repeated for every day where there is an estimate of CTP and HI. Since the process for determining the daily coupling regime does not require the SM variable, the coupling timeseries can extend beyond the availability of SM data if there are CTP and HI data. Therefore, even though the CTP-HI space was classified on data from 2015-2022, the time series of daily coupling was extended to 2003 based on the availability of CTP and HI data from remote sensing."

**Time of CTP-HI "Measurements" vs. AIRS (375-381):** I think that some of this discussion about the time taken from the reanalysis datasets for CTP and HI metrics should go earlier in Methods (Section 3.1). Given that readers are told the AIRS instrument overpass is for 1:30 AM local time, this naturally raises questions about when CTP and HI are being evaluated since it wouldn't make sense to analyze CTP and HI at 1:30 AM. It would be better to not have to wait so long to have those questions answered and mention the sunrise time earlier.

Thank you for your insightful comment. We have revised Section 3 to clarify the timing of the CTP-HI measurements and the rationale for using the AIRS overpass time for validation and the sunrise overpass time for coupling classification. This information is now placed earlier in Methods (Section 3.1).

Revision:
Lines (226-233) - "The merged product is validated using the AIRS overpass time (~1:30 AM local time) to leverage the benefits of remote sensing data (i.e., global coverage). However, the theoretical basis for the CTP-HI framework relies on early morning observations of CTP and HI (Findell and Eltair, 2003; Roundy et al. 2013), which more closely align with the SMAP overpass time (~6 AM local time). Estimates of CTP and HI calculated from reanalysis at 1:30 AM and 6:30 AM reveal variations that suggest that the timing of data acquisition may influence these measurements. Therefore, the merged product is created at two different times, the AIRS overpass time (~1:30AM) and at sunrise. The validation of the merged product is based on CTP and HI calculated at the AIRS overpass time so that it can be directly compared to AIRS, while the merged sunrise CTP and HI is used for the analysis on coupling strength to be consistent with previous L-A coupling work."

3. **Defining Persistency (L125-126):** The quantity "persistency" is central to the study but, since its only described conceptually, it can be hard to understand and remember what exactly it represents and how it is computed. Given that the quantity is not defined with an equation, it maybe be helpful to readers to refer to it as a probability (if I'm understanding it correctly) in some of the figures and when it is first discussed in the results. Perhaps representing it as a probability (%) in Figure 6 could make that more clear?

Thank you for your insightful comment. We recognize that our initial explanation of coupling strength was insufficient. Therefore, we have revised Section 2.2, now section 2.1.3 to ensure that the concept of coupling strength is clearly explained.

Lines (151-159): - "The coupling strength is calculated by applying a first order three-state (dry, wet, and atmospherically controlled) Markov Chain model to the timeseries. A first order three state Markov Chain describes the evolution of the coupling state through three persistence probabilities and six transitional probabilities based on a one-day lag (i.e. tomorrows coupling state is only dependent on the current coupling state). Of the nine probabilities calculated, only the two persistence probabilities for the dry and wet regimes are used to define the coupling strength (i.e. the probability that it remains in its current state). Since coupling strength is a probability expressed as a percentage (ranging from 0% to 100%, with 100% indicating strong coupling or persistence), higher percentages signify a stable interaction between the land and the atmosphere that can impact weather patterns and short-term climate variability. In contrast, low values of coupling strength indicate weaker L-A interactions."

As suggested, we also added (%) to all the figures that showed coupling strength to remind the reader that coupling strength is a probability as described in section 2.1.3. This included revising Figures 6 (line : 457), 8 (line : 504),and 9 (line : 517).

**Minor Edits**

1.  Figure 3a: Not sure having all the scatterplots displayed here is useful, given that there are so many and requires careful consideration of the axes. Perhaps this figure would be more readable as a table?

    We understand your concern about the readability of the scatterplots. The scatterplots provide a visual representation of the correlated errors between the datasets across the 529 global sites, offering insights into relationships and correlations that might be less intuitive in a tabular format. To improve readability, we revised the scatterplots to box plots, which will make the information clearer and easier to interpret.
    Revision:
    Line (349)

[Figure]

2. Line 36: Maybe a citation for ECV for those who are unfamiliar?

We have added an appropriate citation to provide context for readers who may be unfamiliar with Essential Climate Variables.

Revision:

Lines (33-35)

It serves as a vital component in the climate system and represents an essential climate variable (ECV) (Liu et al., 2020; Miranda Espinosa et al., 2020; Pratola et al., 2015).

Overall, this was a well-performed study, and I am looking forward to the publication of this work. Some minor improvements on the readability of the manuscript would go a long way in helping the paper better reach its intended audience.

Thank you for your feedback and encouragement. We appreciate your acknowledgment of the study's quality and your anticipation of its publication. We believe these changes will help the paper better reach and engage its intended audience. Thank you again for your valuable input.